# Selective knowledge sharing for privacy-preserving federated distillation without a good teacher

Jiawei Shao [1], Fangzhao Wu [2] & Jun Zhang [1]

While federated learning (FL) is promising for efficient collaborative learning without revealing local data, it remains vulnerable to white-box privacy attacks, suffers from high communication overhead, and struggles to adapt to heterogeneous models. Federated distillation (FD) emerges as an alternative paradigm to tackle these challenges, which transfers knowledge among clients instead of model parameters. Nevertheless, challenges arise due to variations in local data distributions and the absence of a well-trained teacher model, which leads to misleading and ambiguous knowledge sharing that significantly degrades model performance. To address these issues, this paper proposes a *selective knowledge sharing* mechanism for FD, termed *Selective-FD*, to identify accurate and precise knowledge from local and ensemble predictions, respectively. Empirical studies, backed by theoretical insights, demonstrate that our approach enhances the generalization capabilities of the FD framework and consistently outperforms baseline methods. We anticipate our study to enable a privacy-preserving, communication-efficient, and heterogeneity-adaptive federated training framework.

The rapid development of deep learning (DL)[1] has paved the way for its widespread adoption across various application domains, including medical image processing[2], intelligent healthcare[3], and robotics[4]. The key ingredients of DL include massive datasets and powerful computing platforms, which makes centralized training a typical approach for building high-performing models. However, regulations such as the General Data Protection Regulation (GDPR)[5] and California Consumer Privacy Act (CCPA)[6] have been implemented to limit data collection and storage since the data may contain sensitive personal information. For instance, collecting chest X-ray images from multiple hospitals to curate a large dataset for pneumonia detection is practically difficult, since it would violate patient privacy laws and regulations such as Health Insurance Portability and Accountability Act (HIPAA)[7]. While these restrictions as well as regulations are essential for privacy protection, they hinder the utilization of centralized training in practice. Meanwhile, many domains face the "data islands" problem. For instance, individual hospitals may possess only a limited number of data samples for rare diseases, which makes it difficult to develop accurate and robust models.

Federated learning (FL)[8–10] is a promising technique that can effectively utilize distributed data while preserving privacy. In particular, multiple data-owning clients collaboratively train a DL model by updating models locally on private data and aggregating them globally. These two steps iterate many times until convergence, while private data is kept local. Despite many benefits, FL faces challenges and poses inconveniences. Specifically, the periodical model exchange in FL entails communication overhead that scales up with the model size. This prohibits the use of large-sized models in FL[11,12], which severely limits the model accuracy. Besides, standard federated training methods enforce local models to adopt the same architecture, which cannot adapt well to heterogeneous clients equipped with different computation resources[13,14]. Furthermore, although the raw data are not directly shared among clients, the model parameters may encode private information about datasets.

[1]Hong Kong University of Science and Technology, Hong Kong, China. [2]Microsoft Research Asia, Beijing, China. ✉e-mail: wufangzhao@gmail.com; eejzhang@ust.hk

This makes the shared models vulnerable to white-box privacy attacks[15,16].

To resolve the above difficulties of FL requires us to rethink the fundamental problem of privacy-preserving collaborative learning, which is to effectively share *knowledge* among distributed clients while preserving privacy. Knowledge distillation (KD)[17,18] is an effective technique for transferring knowledge from well-trained teacher models to student models by leveraging proxy samples. The inference results by the teacher models on the proxy samples represent *privileged* knowledge, which supervises the training of the student models. In this way, high-quality student models can be obtained without accessing the training data of the teacher models. Applying KD to collaborative learning gives rise to a paradigm called federated distillation (FD)[19–22], where the ensemble of clients' local predictions on the proxy samples serves as privileged knowledge. By sharing the hard labels (i.e., predicted results)[23] of proxy samples instead of model parameters, the FD framework largely reduces the communication overhead, can support heterogeneous local models, and is free from white-box privacy attacks. Nevertheless, without a well-trained teacher, FD relies on the ensemble of local predictors for distillation, making it sensitive to the training state of local models, which may suffer from poor quality and underfitting. Besides, the non-identically independently distributed (non-IID) data distributions[24,25] across clients exacerbate this issue, since the local models cannot output accurate predictions on the proxy samples that are outside their local distributions[26]. To address the negative impact of misleading knowledge, an alternative is to incorporate soft labels (i.e., normalized logits)[17] during knowledge distillation to enhance the generalization performance. Soft labels provide rich information about the relative similarity between different classes, enabling student models to generalize effectively to unseen examples. However, a previous study[20] pointed out that ensemble predictions may be ambiguous and exhibit high entropy when local predictions of clients are inconsistent. Sharing soft labels can exacerbate this problem as they are less certain than hard labels. The misleadingness and ambiguity harm the knowledge distillation and local training. For instance, in our experiments on image classification tasks, existing FD methods barely outperform random guessing under highly non-IID distributions.

This work aims to tackle the challenge of knowledge sharing in FD without a good teacher, and our key idea is to filter out misleading and ambiguous knowledge. We propose a *selective knowledge sharing*

mechanism in federated distillation (named *Selective-FD*) to identify accurate and precise knowledge during the federated training process. As shown in Figs. 1 and 2, this mechanism comprises client-side selectors and a server-side selector. At each client, we construct a selector to identify out-of-distribution (OOD) samples[27,28] from the proxy dataset based on the density-ratio estimation[29]. This method detects outliers by quantifying the difference in densities between the inlier distribution and outlier distribution. If the density ratio of a particular sample is close to zero, the client considers it an outlier and refrains from sharing the predicted result to prevent misleading other clients. On the server side, we average the uploaded predictions from the clients and filter out the ensemble predictions with high entropy. The other ensemble predictions are then returned to the clients for knowledge distillation. We provide theoretical insights to demonstrate the impact of our selective knowledge sharing mechanism on the training convergence, and we evaluate Selective-FD in two applications, including a pneumonia detection task and three benchmark image classification tasks. Extensive experimental results show that Selective-FD excels in handling non-IID data and significantly improves test accuracy compared to the baselines. Remarkably, Selective-FD with hard labels achieves performance close to the one sharing soft labels. Furthermore, our proposed Selective-FD significantly reduces the communication cost during federated training compared with the conventional FedAvg approach. We anticipate that the proposed method will serve as a valuable tool for training large models in the federated setting for future applications.

## Results
### Performance evaluation

The experiments are conducted on a pneumonia detection task[30] and three benchmark image classification tasks[31–33]. The pneumonia detection task aims to detect pneumonia from chest X-ray images. This task is based on the COVIDx dataset[30] that contains three classes, including normal people, non-COVID-19 infection, and COVID-19 viral infection. We consider four clients, e.g., hospitals, in the federated distillation framework. To simulate the non-IID data across clients, each of them only has one or two classes of chest X-ray images, and each class contains 1,000 images. Besides, we construct a proxy dataset for knowledge transfer, which contains all three classes, and each class has 500 unlabeled samples. The non-IID data distribution is visualized in Fig. 3. The test dataset consists of 100 normal images and

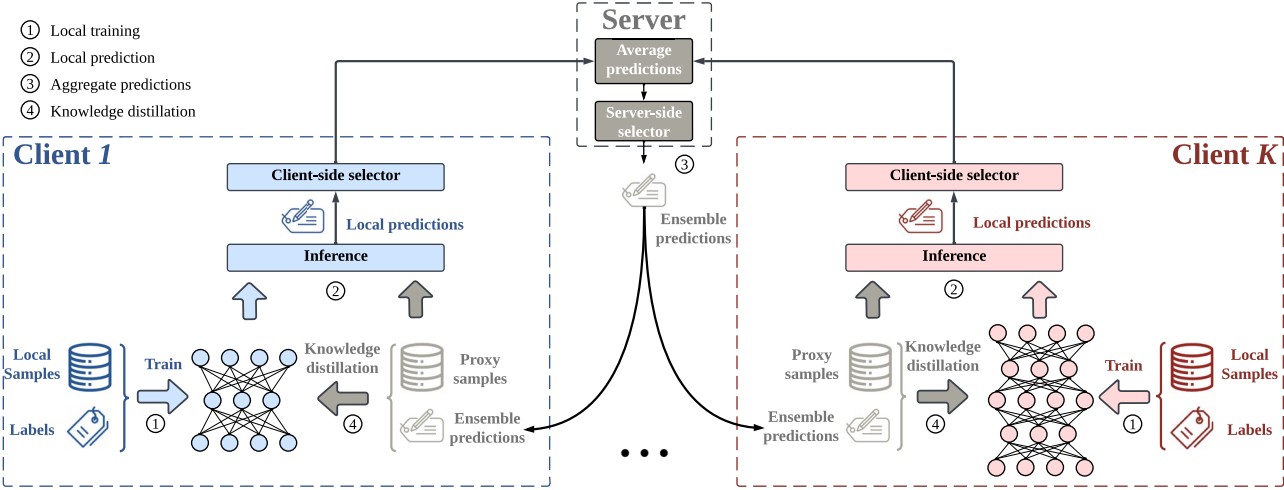

**Fig. 1 | The overall framework of Selective-FD.** The federated distillation involves four iterative steps. First, each client trains a personalized model using its local private data. Second, each client predicts the label of the proxy samples based on the local model. Third, the server aggregates these local predictions and returns the ensemble predictions to clients. Fourth, clients update local models by knowledge

distillation based on the ensemble predictions. During the training process, the client-side selectors and the server-side selector aim to filter out misleading and ambiguous knowledge from the local predictions. Some icons in this figure are from icons8.com.

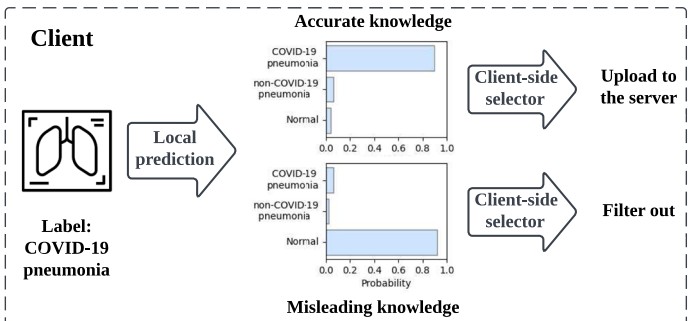
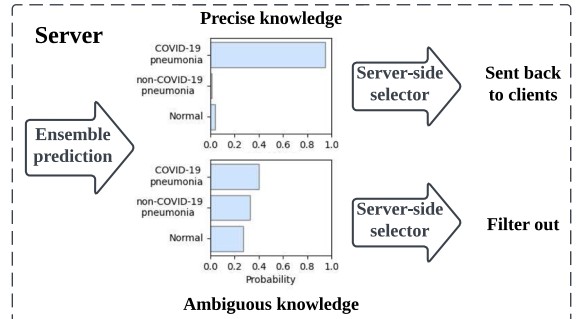

**Fig. 2 | Predict the label of a proxy sample on the client side (left) and the server side (right).** The prediction takes the form of a soft label (i.e., a logits vector), where each element represents the probability of the corresponding label. The predicted label is the element with the highest probability. We measure the quality of knowledge in federated distillation by accuracy and precision. The accurate prediction matches the ground truth label, while misleading knowledge does not.

Meanwhile, precise knowledge has low entropy, while ambiguity implies high entropy and uncertainty. The client-side selectors are responsible for filtering out incorrect local predictions, while the server-side selector aims to eliminate ambiguous knowledge. The X-ray icon in this figure is from Chanut-is-Industries, Flaticon.com.

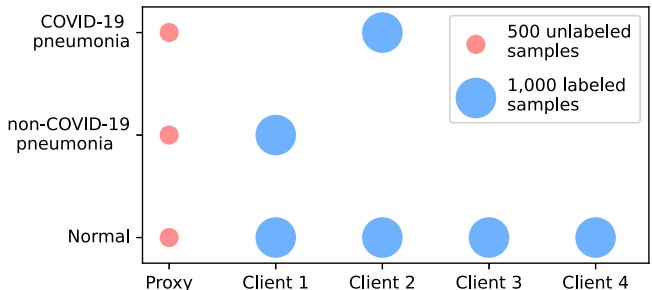

**Fig. 3 | Visualization of non-IID data distribution.** The horizontal axis indexes the proxy dataset and the local datasets, while the vertical axis indicates class labels. The size of scattered points denotes the number of samples.

100 images of pneumonia infection, where half of the pneumonia infections are non-COVID-19 infections and the other half are COVID-19 infections. Moreover, we evaluate the proposed method on three benchmark image datasets, including MNIST[31], Fashion MNIST[32], and CIFAR-10[33]. The datasets consist of ten classes, each with over 50,000 training samples. To transfer knowledge in a federated distillation setting, we randomly select 10% to 20% of the training data from each class as unlabeled proxy samples. In the experiments, ten clients participate in the distillation process, and we evaluate the model's performance under two non-IID distribution settings: a strong non-IID setting and a weak non-IID setting, where each client has one unique class and two classes, respectively. Several representative federated distillation methods are compared, including FedMD[13], FedED[19], DS-FL[20], FKD[34], and PLS[26]. Among them, FedMD, FedED, and DS-FL rely on a proxy dataset to transfer knowledge, while FKD and PLS are data-free KD approaches that share class-wise average predictions among users. Besides, we report the performance of independent learning (abbreviated as IndepLearn), where each client trains the model on its local dataset. The comparison includes the results of sharing hard labels (predicted labels) and soft labels (normalized logits). To evaluate the training performance of these methods, we report the classification accuracy on the test set as a metric. More details of the datasets and model structures are deferred to Supplementary Information.

The average test accuracy of clients on the pneumonia detection task is depicted in Fig. 4. It is observed that the proposed Selective-FD method outperforms all the baseline methods by a substantial margin, and the performance gain is more significant when using hard labels to transfer knowledge. For example, sharing knowledge by hard labels and soft labels resulted in improvements of 19.42% and 4.00%,

respectively, over the best-performed baseline. This is because the proposed knowledge selection mechanism can adapt to the heterogeneous characteristics of local data, making it effective in selecting useful knowledge among clients. In contrast, some baselines perform even worse than the independent learning scheme. This finding highlights the potential negative influence of knowledge sharing among clients, which can mislead the local training. Notably, while hard label sharing provides a stronger privacy guarantee[35], soft label sharing provides additional performance gains. This is because soft labels provide more information about the relationships between classes than hard labels, alleviating errors from misleading knowledge.

We also evaluate the performance of different federated distillation methods on the benchmark image datasets. As shown in Table 1, we find that all the methods achieve satisfactory accuracy in the IID setting. On the contrary, the proposed Selective-FD method consistently outperforms the baselines when local datasets are heterogeneous. The improvement of our method becomes more significant as the severity of the non-IID problem increases. Specifically, it is observed that the FKD and PLS methods degrade to IndepLearn in the strong non-IID setting. This is because each client only possesses one unique class, and the local predictions are always that unique class. Such misleading knowledge leads to significant performance degradation.

## Effectiveness of density-ratio estimation

We verify the effectiveness of the density-ratio estimation in detecting incorrect predictions of local models. Specifically, as an ablation study, we replace the density-ratio based selector in Selective-FD with confidence-based methods[27] and energy-based models (EBMs)[28], respectively. The confidence score refers to the maximum probability of the logits in the classification task, which reflects the reliability of the prediction. The EBMs distinguish the in-distribution samples and out-distribution samples by learning an underlying probability distribution over the training samples. The predictions of proxy samples detected as out-distribution samples will be ignored.

Our experiments are conducted on benchmark datasets under the strong non-IID setting, where hard labels are shared among clients for distillation. We use the area under the receiver operating characteristic (AUROC) metric to measure the capability of selectors to detect incorrect predictions. In addition, we evaluate the performance of various selectors by reporting test accuracy. As shown on the left-hand side of Fig. 5, the AUROC score of our method is much higher than the baselines. Particularly, the confidence-based method and energy-based model perform only marginally better than random guess (AUROC = 0.5) on the MNIST and Fashion MNIST datasets. This is

because the neural networks tend to be over-confident[36] about the predictions, and thus the confidence score may not be able to reflect an accurate probability of correctness for any of its predictions. Besides, the energy-based model fails to detect the incorrect predictions because they often suffer from the problem of overfitting without the out-of-distribution samples[37]. The right-hand side of Fig. 5 shows the test accuracy after federated distillation. As the density-ratio estimation can effectively identify unknown classes from the proxy samples, the ensemble knowledge is less misleading and our Selective-FD approach achieves a significant performance gain.

### Ablation study on thresholds of selectors

In Selective-FD, the client-side selectors and the server-side selector are designed to remove misleading and ambiguous knowledge, respectively. Two important parameters are the thresholds $\tau_{\text{client}}$ and $\tau_{\text{server}}$. Specifically, each client reserves a portion of the local data as a validation set. The threshold of the client-side selector is defined as the $\tau_{\text{client}}$ quantile of the estimated ratio over this set. When the density ratio of a sample falls below this threshold, the respective prediction is considered to be misleading. Besides, the server-side selector filters out the ambiguous knowledge according to the confidence score of the ensemble prediction. Specifically, when a confidence score is smaller than $1 - \tau_{\text{server}}/2$, the corresponding proxy sample will not be used for knowledge distillation.

We investigate the effect of the thresholds $\tau_{\text{client}}$ and $\tau_{\text{server}}$ on the performance. We conduct experiments on three benchmark image datasets in the strong non-IID setting, where the predictions shared among clients are soft labels. Fig. 6 displays the results, with $p_{\text{proxy}}$ representing the percentage of proxy samples used to transfer knowledge during the whole training process. The threshold $\tau_{\text{client}}$ is set as 0.25 when evaluating the performance of $\tau_{\text{server}}$, and the threshold $\tau_{\text{server}}$ is set as 2 when evaluating the performance of $\tau_{\text{client}}$. We have observed that both $\tau_{\text{client}}$ and $\tau_{\text{server}}$ have a considerable impact on the performance. When $\tau_{\text{client}}$ is set too high or $\tau_{\text{server}}$ is set too low, a significant number of proxy samples are filtered out by the server-side selector, which decreases the test accuracy. On the other hand, setting $\tau_{\text{client}}$ too low may cause the client-side selectors to be unable to remove the inaccurate local predictions, leading to a negative impact on knowledge distillation. When $\tau_{\text{server}}$ is set too high, the server-side selector fails to identify the ensemble predictions with high entropy, which results in a drop in accuracy. These empirical results align with Theorem 2 and the analysis presented in Remark 1.

### Comparison with FedAvg

Compared with the standard FL setting, such as FedAvg[8], Selective-FD introduces a different approach by sharing knowledge instead of model parameters during the training process. This alternative method offers several advantages. First, Selective-FD naturally adapts to heterogeneous models, eliminating the need for local models to share the same architecture. Second, Selective-FD largely reduces the communication overhead in comparison to FedAvg, since the size of knowledge is significantly smaller than the model. Third, Selective-FD provides a stronger privacy guarantee than FedAvg. The local models, which might contain encoded information from private datasets[22], remain inaccessible to other clients or the server. To better demonstrate the advantages in communication efficiency and privacy protection offered by the proposed method, the following content provides the quantitative comparisons between Selective-FD and FedAvg.

**Communication overhead.** We compare the communication overhead of Selective-FD with FedAvg on the benchmark datasets in the strong non-IID setting. In each communication round of FedAvg, the clients train their models locally and upload them to the server for aggregation. This requires that all local models have the same architecture. For the MNIST classification task, the local models consist of two convolutional layers and two fully-connected layers. In the case of Fashion MNIST, we initialize each model as a Multilayer Perceptron (MLP) with two hidden layers, each containing 1024 neurons. Furthermore, we employ ResNet18[38] as the local model to classify CIFAR-10 images. Our Selective-FD method requires clients to collect proxy samples prior to the training process, which consequently results in an additional communication overhead. However, Selective-FD significantly reduces the amount of data uploaded and downloaded per communication round compared with FedAvg. This is because the size of predictions used for knowledge distillation is much smaller than that of model updates utilized for aggregation. Fig. 7 plots the test accuracy and communication overhead with respect to the communication round.

It is observed that our Selective-FD method has comparable or inferior accuracy to FedAvg. But it considerably improves communication efficiency during federated training. Further improving the performance of Selective-FD is a promising direction for future research. Additional information regarding the experiments can be found in Supplementary Information.

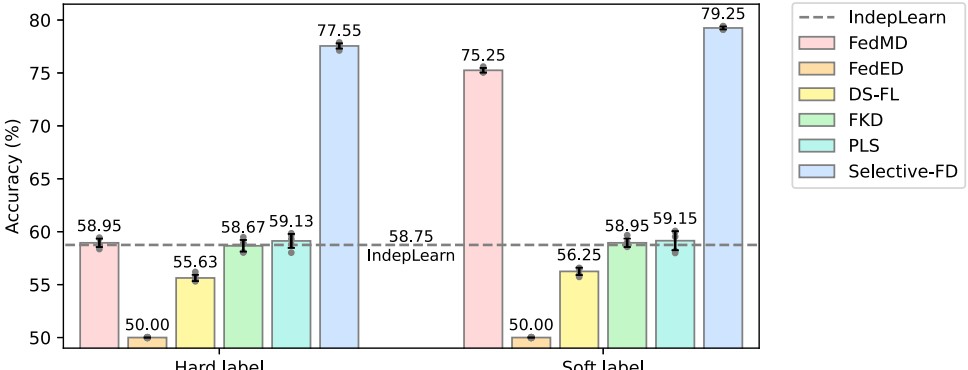

**Fig. 4 | Test accuracy of different methods on the pneumonia detection task.** The error bar represents the mean ± standard deviation of five repetitions. The results show that the proposed Selective-FD method achieves the best performance, and the accuracy gain is more significant when using hard labels to transfer knowledge. Specifically, some baselines perform even worse than the independent learning scheme. These results demonstrate that knowledge sharing among clients can mislead and negatively influence local training. The NIH Chest X-ray Dataset[53] and RSNA-STR Pneumonia Detection Challenge image datasets[54] are used for model training and testing. NIH Chest X-ray Dataset is available at https://nihcc.app.box.com/v/ChestXray-NIHCC, which is provided by the NIH Clinical Center. The RSNA-STR Pneumonia Detection Challenge image datasets is available at https://www.rsna.org/education/ai-resources-and-training/ai-image-challenge/RSNA-Pneumonia-Detection-Challenge-2018.

## Table 1 | Test accuracy of different methods

| Strong non-IID | MNIST | | FashionMNIST | | CIFAR-10 | |
|---|---|---|---|---|---|---|
| | Hard label | Soft label | Hard label | Soft label | Hard label | Soft label |
| IndepLearn | 10.00 ± 0.00 | | 10.00 ± 0.00 | | 10.00 ± 0.00 | |
| FedMD | 18.89 ± 0.30 | 88.71 ± 0.28 | 16.54 ± 0.25 | 64.63 ± 0.37 | 10.71 ± 0.38 | 15.78 ± 1.39 |
| FedED | 11.49 ± 0.25 | 11.92 ± 0.41 | 12.45 ± 0.44 | 12.52 ± 0.38 | 11.83 ± 0.26 | 12.04 ± 0.30 |
| DS-FL | 19.72 ± 0.32 | 35.25 ± 0.36 | 17.54 ± 0.11 | 35.98 ± 0.43 | 10.87 ± 0.25 | 12.07 ± 0.32 |
| FKD | 10.00 ± 0.00 | 10.00 ± 0.00 | 10.00 ± 0.00 | 10.00 ± 0.00 | 10.00 ± 0.00 | 10.00 ± 0.00 |
| PLS | 10.00 ± 0.00 | 10.00 ± 0.00 | 10.00 ± 0.00 | 10.00 ± 0.00 | 10.00 ± 0.00 | 10.00 ± 0.00 |
| Selective-FD | **85.92 ± 0.37** | **94.68 ± 0.52** | **73.41 ± 0.98** | **75.31 ± 0.29** | **80.22 ± 0.74** | **80.98 ± 0.39** |
| **Weak Non-IID** | MNIST | | FashionMNIST | | CIFAR-10 | |
| | Hard label | Soft label | Hard label | Soft label | Hard label | Soft label |
| IndepLearn | 19.96 ± 0.00 | | 19.82 ± 0.01 | | 19.52 ± 0.02 | |
| FedMD | 26.77 ± 0.57 | 95.16 ± 0.52 | 41.92 ± 0.30 | 74.83 ± 0.41 | 62.14 ± 0.22 | 84.31 ± 0.53 |
| FedED | 59.95 ± 1.11 | 60.26 ± 1.80 | 32.62 ± 1.09 | 37.12 ± 0.85 | 53.11 ± 0.34 | 56.13 ± 0.14 |
| DS-FL | 25.53 ± 1.43 | 47.87 ± 0.31 | 23.08 ± 0.23 | 39.22 ± 0.26 | 33.22 ± 0.54 | 52.51 ± 0.70 |
| FKD | 19.97 ± 0.01 | 19.98 ± 0.00 | 19.54 ± 0.13 | 19.71 ± 0.07 | 19.50 ± 0.02 | 19.51 ± 0.01 |
| PLS | 19.96 ± 0.01 | 19.97 ± 0.00 | 19.64 ± 0.09 | 19.70 ± 0.03 | 19.51 ± 0.01 | 19.52 ± 0.01 |
| Selective-FD | **86.82 ± 0.26** | **96.30 ± 0.25** | **75.57 ± 0.61** | **77.27 ± 0.31** | **81.06 ± 0.67** | **85.38 ± 0.35** |
| **IID** | MNIST | | FashionMNIST | | CIFAR-10 | |
| | Hard label | Soft label | Hard label | Soft label | Hard label | Soft label |
| IndepLearn | 98.18 ± 0.04 | | 86.07 ± 0.07 | | 84.06 ± 0.03 | |
| FedMD | **98.59 ± 0.04** | **98.63 ± 0.01** | 86.88 ± 0.02 | 87.25 ± 0.02 | 86.02 ± 0.09 | 86.31 ± 0.06 |
| FedED | 98.20 ± 0.11 | 98.26 ± 0.06 | 86.83 ± 0.14 | 86.88 ± 0.04 | **86.54 ± 0.07** | **86.87 ± 0.12** |
| DS-FL | 98.22 ± 0.14 | 98.56 ± 0.02 | 86.15 ± 0.07 | 86.62 ± 0.09 | 85.75 ± 0.08 | 85.82 ± 0.08 |
| FKD | 98.40 ± 0.05 | 98.44 ± 0.01 | 86.10 ± 0.15 | 86.14 ± 0.06 | 84.03 ± 0.02 | 84.10 ± 0.08 |
| PLS | 98.45 ± 0.02 | 98.48 ± 0.03 | 86.27 ± 0.08 | 86.52 ± 0.05 | 84.60 ± 0.13 | 84.77 ± 0.06 |
| Selective-FD | 98.55 ± 0.01 | 98.60 ± 0.04 | **86.92 ± 0.08** | 87.16 ± 0.06 | 85.94 ± 0.07 | 86.06 ± 0.16 |

Each experiment is repeated five times. The results in bold indicate the best performance, while the results underlined represent the second-best performance. In the non-IID settings, our Selective-FD method performs better than the baseline methods, and the accuracy gain is more significant when using hard labels in knowledge distillation than soft labels. In the IID scenario, all the methods achieve satisfactory accuracy.

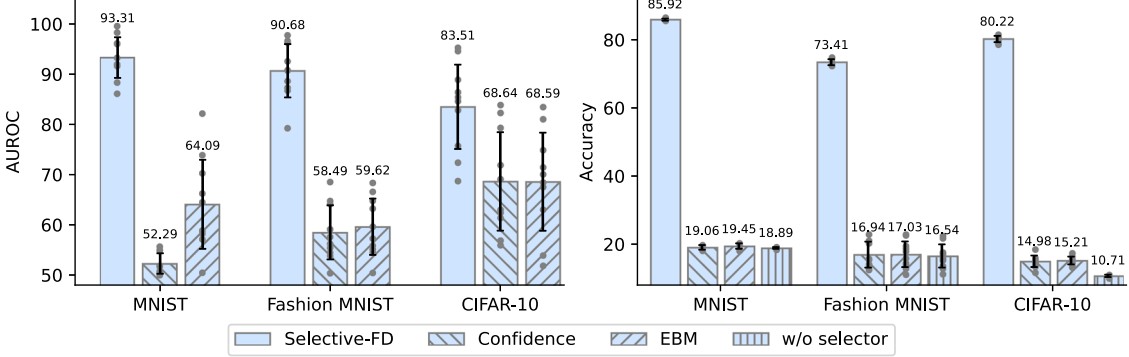

**Fig. 5 | The AUROC scores for incorrect prediction detection (left) and the test accuracy after federated distillation (right).** The error bar represents the mean ± standard deviation across 10 clients. The results show that both the AUROC score and the accuracy of our Selective-FD method are much higher than the baselines, indicating its effectiveness in identifying unknown classes from the proxy dataset. This results in a remarkable performance gain in federated distillation.

**Privacy leakage.** We compare the privacy leakage of Selective-FD and FedAvg under model inversion attacks[18,39]. The objective of the attacker is to reconstruct the private training data based on the shared information from clients. In FedAvg, a semi-honest server can perform white-box attacks[39] based on the model updates. In contrast, our Selective-FD method is free from such attacks since the clients' models cannot be accessed by the server. However, our method remains vulnerable to black-box attacks, where the attacker can infer local samples by querying the clients' models[18]. To assess the privacy risk quantitatively, we employ GMI[39] and IDEAL[18] to attack FedAvg and Selective-FD, respectively. This experiment is conducted on MNIST, and the results

are shown in Fig. 8. It is observed that the quality of reconstructed images from FedAvg is better than that from Selective-FD. This demonstrates that sharing model parameters leads to higher privacy leakage than sharing knowledge. Besides, compared with sharing soft labels in Selective-FD, the reconstructed images inferred from the hard labels have a lower PSNR value. This indicates that sharing hard labels in Selective-FD exposes less private information than sharing soft labels. This result is consistent with Hinton's analysis[40], where the soft labels provide more information per training case. In federated training where the local data are privacy-sensitive, such as large genomic datasets[41], it becomes crucial to share hard labels rather than soft

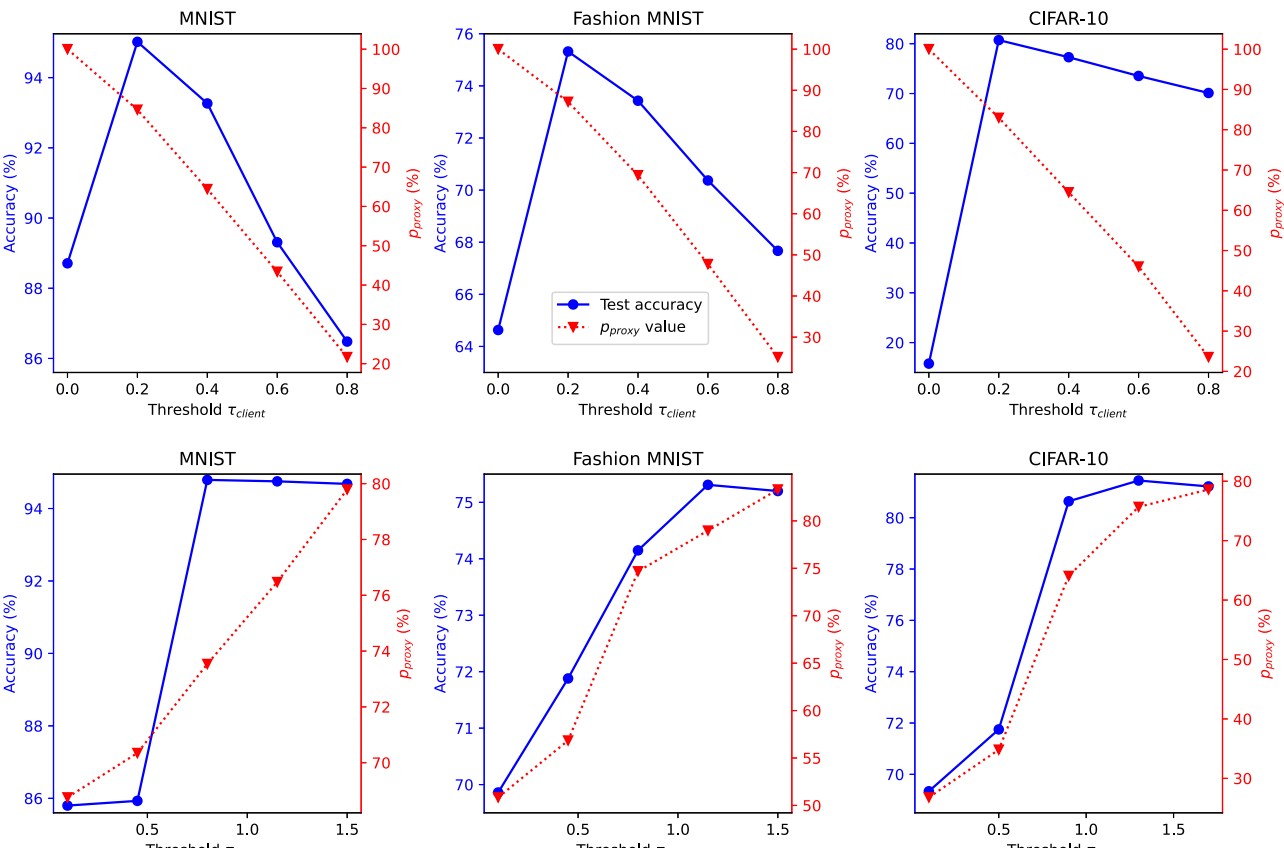

**Fig. 6 | Test accuracy and percentage $p_{proxy}$ under (top) different values of $\tau_{client}$ and (bottom) different values of $\tau_{server}$.** We denote the percentage of proxy samples selected for knowledge distillation as $p_{proxy}$. When $\tau_{client}$ is too large or $\tau_{server}$ is too small, the selectors filter out most of the proxy samples, leading to a small batch size and increased training variance. Conversely, when $\tau_{client}$ is too small, the local outputs may contain an excessive number of incorrect predictions, leading to a decrease in the effectiveness of knowledge distillation. Besides, when $\tau_{server}$ is too large, the ensemble predictions may exhibit high entropy, indicating ambiguous knowledge that could degrade local model training. These empirical results align with the analysis in Remark 1.

labels. This serves as a protective measure against potential membership inference attacks[42]. Finally, although the knowledge sharing methods provide stronger privacy guarantees compared with FedAvg, the malicious attackers can still infer the label distribution of clients from the shared information. Developing a privacy-enhancing federated training scheme is a promising but challenging direction for future research.

## Discussion

This work introduces the Selective-FD method in federated distillation, which includes a selective knowledge sharing mechanism that identifies accurate and precise knowledge from clients for effective knowledge transfer. Particularly, it includes client-side selectors and a server-side selector. The client-side selectors use density-ratio estimators to identify out-of-distribution samples from the proxy dataset. If a sample exhibits a density ratio that approaches zero, it is identified as an outlier. To prevent the propagation of potentially misleading information to other clients, this sample is not used for knowledge distillation. Besides, as the local models could be underfitting at the beginning of the training process, the local predictions could be inconsistent among clients. To prevent the negative influence of ambiguous knowledge, the server-side selector filters out the ensemble predictions with high entropy.

Extensive experiments are conducted on both pneumonia detection and benchmark image classification tasks to investigate the impact of hard labels and soft labels on the performance of knowledge distillation. The results demonstrate that Selective-FD significantly improves test accuracy compared to FD baselines, and the accuracy gain is more prominent in hard label sharing than in soft label sharing. In comparison with the standard FL framework that shares model parameters among clients, sharing knowledge in FD may not achieve the same performance level, but this line of work is communication-efficient, privacy-preserving, and heterogeneity-adaptive. When performing federated training on large language models (LLM)[43,44], the FD framework is especially useful since the clients do not need to upload the huge amount of model parameters, and it is difficult for attackers to infer the private texts. We envision that our proposed method can serve as a template framework for various applications and inspire future research to improve the effectiveness and responsibility of intelligence systems.

However, we must acknowledge that our proposed method is not without its limitations. Firstly, the federated distillation method relies on proxy samples to transfer knowledge. If the proxy dataset is biased towards certain classes, the client models may be biased toward these classes. This leads to poor performance and generalization. Secondly, the complexity of the client-side selector in our Selective-FD method increases quadratically with the number of samples and the sample space, which may limit its practical applicability. Some studies in open-set learning have shown that other low-complexity outlier detection methods, while lacking theoretical guarantees, can achieve comparable performance. This finding motivates us to explore more efficient selectors in future research. Thirdly, while our proposed method keeps models local, it cannot guarantee perfect privacy. Attackers may infer information from the private dataset based on the shared knowledge. To further strengthen privacy guarantees in FD, we could employ defense methods such as differential privacy mechanisms and secure

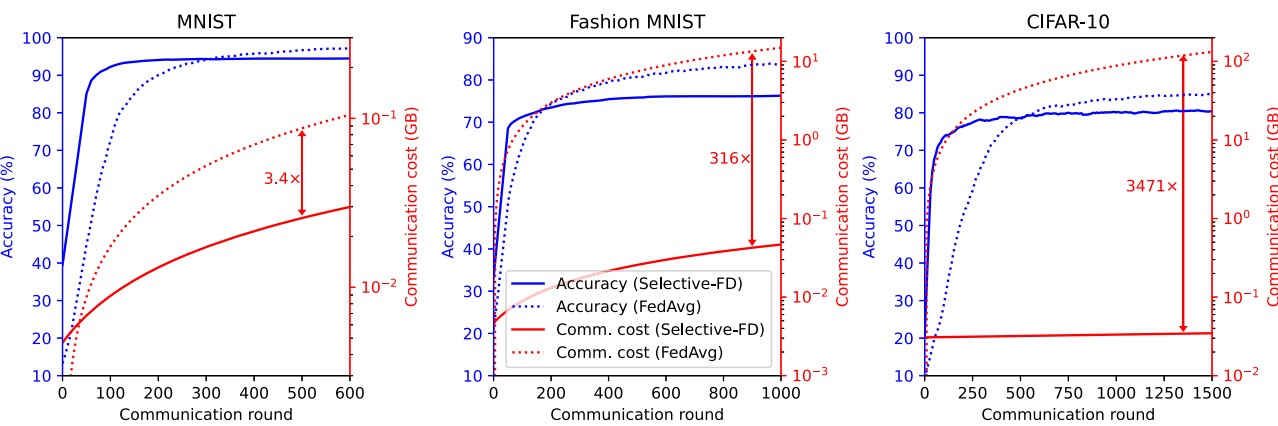

**Fig. 7 | Test accuracy and communication cost as functions of the communication round.** Our Selective-FD method has a comparable accuracy as FedAvg in the MNIST and CIFAR-10 datasets but is inferior in Fashion MNIST.

However, Selective-FD achieves a significant reduction in the communication overhead, which is because the cost of model sharing in FedAvg is much higher than knowledge sharing in our method.

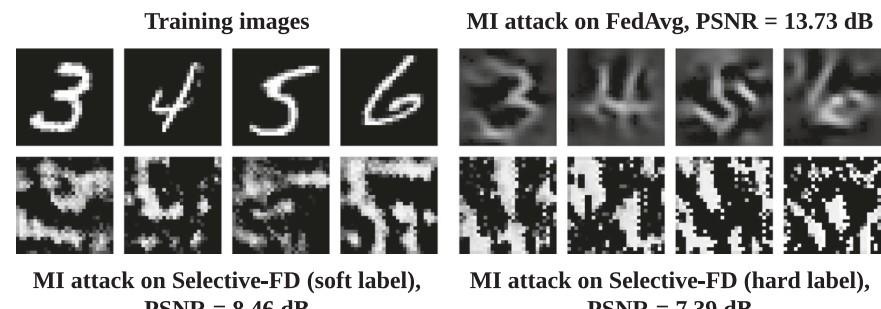

**Fig. 8 | Visualization of the reconstructed images by performing model inversion (MI) attacks.** We utilize the Peak signal-to-noise ratio (PSNR) as a metric to quantitatively evaluate the quality of the reconstructed images. A higher PSNR value indicates an increase in privacy risk.

aggregation protocols. However, these defense methods might lead to performance degradation or increased training complexity. Future research should develop more efficient and effective defensive techniques to protect clients against attacks while maintaining optimal performance.

### Ethical and societal impact
It is essential to note that the proposed federated distillation framework, like other AI algorithms, is a dual-use technology. Although the objective is to facilitate the training process of deep learning models by effectively sharing knowledge among clients, this technology can also be misused. Firstly, federated distillation could be used to train algorithms among malicious entities to identify individuals illegally and monitor the speaking patterns of people without their consent[45]. This poses threats to individual freedom and civil liberties. Secondly, it is assumed that all the participants in federated distillation, including the clients and the server, are trusted and behave honestly. However, if a group of clients or the server are malicious, they could manipulate the predictions and add poison knowledge during the training process, heavily degrading the convergence[46,47]. Thirdly, while federated distillation is free from white-box attacks, it is still potentially vulnerable to black-box attacks such as membership inference attacks[35]. This is because the local predictions are shared among clients, which may allow an attacker to infer whether a particular sample is part of a client's training set or not. To mitigate this vulnerability, additional privacy-preserving techniques such as differential privacy or secure aggregation can be employed. Furthermore, there are concerns about the collection of proxy samples to transfer knowledge between clients. This could potentially lead to breaches of privacy and security, as well as ethical concerns regarding informed consent and data ownership. In summary,

while the proposed federated distillation framework has great potential for facilitating collaborative learning among multiple clients, it is important to be aware of the potential risks and take measures to ensure that the technology is used ethically and responsibly.

## Methods
In this section, we provide an in-depth introduction to our proposed method. We first define the problem studied in this paper, then introduce the details of our method, and provide theoretical insights to discuss the impact of the selective knowledge sharing mechanism on generalization.

### Notations and problem definition
Consider a federated setting for multi-class classification where each input instance has one ground truth label from $C$ categories. We index clients as $1, 2, \ldots, K$, where the $k$-th client stores a private dataset $\hat{\mathcal{D}}_k$, consisting of $m_k$ data points sampled from the distribution $\mathcal{D}_k$. In addition, there is a proxy dataset $\hat{\mathcal{D}}_{\text{proxy}}$ containing $m_{\text{proxy}}$ samples from the distribution $\mathcal{D}_{\text{proxy}}$ to transfer knowledge among clients. We denote $\mathcal{X}$ as the input space and $\mathcal{Y} \in V(\Delta^{C-1})$ as the label space. Specifically, $V(\Delta^{C-1})$ is a vertex set consisting of all vertices in a $C-1$ simplex $\Delta^{C-1}$, where each vertex corresponds to a one-hot vector. We assume that all the local datasets and the test dataset have the same labeling function $\hat{\boldsymbol{h}}^* : \mathcal{X} \to V(\Delta^{C-1})$. Client $k$ learns a local predictor $\boldsymbol{h}_k : \mathcal{X} \to \Delta^{C-1}$ to approach $\hat{\boldsymbol{h}}^*$ in the training phase and outputs a one-hot vector by $\hat{\boldsymbol{h}}_k : \mathcal{X} \to V(\Delta^{C-1})$ in the test phase. The hypothesis spaces of $\boldsymbol{h}_k$ and $\hat{\boldsymbol{h}}_k$ are $\mathcal{H}_k$ and $\hat{\mathcal{H}}_k$, respectively, which are determined by the parameter spaces of the personalized models. During the process of knowledge

sharing, we define the labeling function of the proxy samples as $\boldsymbol{h}^*_{\text{proxy}} : \mathcal{X} \to \Delta^{C-1}$, which is determined by both the local predictors and the knowledge selection mechanism. To facilitate the theoretical analysis, we denote $\hat{\boldsymbol{h}}^*_{\text{proxy}} : \mathcal{X} \to V(\Delta^{C-1})$ as the one-hot output of $\boldsymbol{h}^*_{\text{proxy}}$.

In the experiments, we use cross entropy as the loss function, while for the sake of mathematical traceability, the $\ell_1$ norm is adopted to measure the difference between two predictors, denoted by $\mathcal{L}_{\mathcal{D}}(\hat{\boldsymbol{h}}, \hat{\boldsymbol{h}}') := \mathbb{E}_{\mathbf{x} \sim \mathcal{D}} \| \hat{\boldsymbol{h}}(\mathbf{x}) - \hat{\boldsymbol{h}}'(\mathbf{x}) \|_1$, where $\mathcal{D}$ is the distribution. Specifically, $\mathcal{L}_{\mathcal{D}_{\text{test}}}(\hat{\boldsymbol{h}}_k, \hat{\boldsymbol{h}}^*)$, $\mathcal{L}_{\hat{\mathcal{D}}_k}(\hat{\boldsymbol{h}}_k, \hat{\boldsymbol{h}}^*)$, and $\mathcal{L}_{\hat{\mathcal{D}}_{\text{proxy}}}(\hat{\boldsymbol{h}}_k, \boldsymbol{h}^*_{\text{proxy}})$ represent the loss over the test distribution, local samples, and the proxy samples, respectively. Besides, we define the training loss over both the private and proxy samples as $\mathcal{L}_{\hat{\mathcal{D}}_k \cup \hat{\mathcal{D}}_{\text{proxy}}}(\hat{\boldsymbol{h}}_k) := \alpha \mathcal{L}_{\hat{\mathcal{D}}_k}(\hat{\boldsymbol{h}}_k, \hat{\boldsymbol{h}}^*) + (1 - \alpha) \mathcal{L}_{\hat{\mathcal{D}}_{\text{proxy}}}(\hat{\boldsymbol{h}}_k, \boldsymbol{h}^*_{\text{proxy}})$, where $\alpha \in [0, 1]$ is a weighted coefficient. The notation $\text{Pr}_{\mathcal{D}}[\cdot]$ represents the probability of events over the distribution $\mathcal{D}$.

## Selective knowledge sharing in federated distillation

Federated distillation aims at collaboratively training models among clients by sharing knowledge, instead of sharing models as in FL. The training process is available in Algorithm 1, which involves two phases. First, the clients train the local models independently on the local data. Then, the clients share knowledge among themselves based on a proxy dataset and fine-tune the local models over both the local and proxy samples. Fig. 1 provides an overview of our Selective-FD framework, which includes a selective knowledge sharing mechanism. The following section will delve into the details of this mechanism.

**Client-side selector**. Federated distillation presents a challenge of misleading ensemble predictions caused by the lack of a well-trained teacher. Local models may overfit the local datasets, leading to poor generalization on proxy samples outside the local distribution, especially with non-IID data across clients. To mitigate this issue, our method develops client-side selectors to identify proxy samples that are out of the local distribution (OOD). This is done through density-ratio estimation[48,49], which calculates the ratio of two probability densities. Assuming the input data space $\mathcal{X}$ is compact, we define $\mathcal{U}$ as a uniform distribution with probability density function $u(\mathbf{x})$ over $\mathcal{X}$. Besides, we denote the probability density function at client $k$ as $p_k(\mathbf{x})$. Our objective is to estimate the density ratio $\boldsymbol{w}^*_k(\mathbf{x}) = p_k(\mathbf{x})/u(\mathbf{x})$ based on the observed samples. Specifically, the in-sampple data $\mathbf{x}$ from the local distribution with $p_k(\mathbf{x}) > 0$ results in $\boldsymbol{w}^*_k(\mathbf{x}) > 0$, while the OOD samples $\mathbf{x}$ with probability $p_k(\mathbf{x}) = 0$ leads to $\boldsymbol{w}^*_k(\mathbf{x}) = 0$. Therefore, the clients can build density-ratio estimators to identify the OOD samples from the proxy dataset.

Considering the property of statistical convergence, we use a kernelized variant of unconstrained least-squares importance fitting (KuLSIF)[29] to estimate the density ratio. The estimation model in KuLSIF is a reproducing kernel Hilbert space (RKHS)[50] $\mathcal{W}_k$ endowed with a Gaussian kernel function. We sample $n_k$ and $n_u$ data points from $\mathcal{D}_k$ and $\mathcal{U}$, respectively, and denote the resulting sample sets as $S_k$ and $S_u$. Defining the norm on $\mathcal{W}_k$ as $\| \cdot \|_{\mathcal{W}_k}$, the density-ratio estimator $\boldsymbol{w}_k$ is obtained as an optimal solution of

$$\boldsymbol{w}_k = \text{argmin}_{\boldsymbol{w}'_k \in \mathcal{W}_k} \frac{1}{2n_u} \sum_{\mathbf{x} \sim S_u} (\boldsymbol{w}'_k(\mathbf{x}))^2 - \frac{1}{n_k} \sum_{\mathbf{x} \sim S_k} \boldsymbol{w}'_k(\mathbf{x}) + \frac{\beta}{2} \| \boldsymbol{w}'_k \|^2_{\mathcal{W}_k}, \quad (1)$$

where the analytic-form solution $\boldsymbol{w}_k$ is available in Theorem 1 of the KuLSIF method[29]. The following theorem reveals the convergence rate of the KuLSIF estimator.

**Theorem 1.** (Convergence rate of KuLSIF[29]). Consider RKHS $\mathcal{W}_k$ to be the Hilbert space with Gaussian kernel that contains the density ratio $\boldsymbol{w}^*_k$. Given $\delta \in (0, 1)$ and setting the regularization $\beta = \beta_{n_k, n_u}$ such that $\lim_{n_k, n_u \to 0} \beta_{n_k, n_u} = 0$ and $\beta^{-1}_{n_k, n_u} = O\left(\min\{n_k, n_u\}^{1-\delta}\right)$, we have $\mathbb{E}_{\mathbf{x} \sim u(\mathbf{x})} \| \boldsymbol{w}_k(\mathbf{x}) - \boldsymbol{w}^*_k(\mathbf{x}) \| = O_p\left(\beta^{1/2}_{n_k, n_u}\right)$, where $O_p$ is the probability order.

The proof is available in Theorem 2 of the KuLSIF method[29]. This theorem demonstrates that as the number of samples increases and the regularization parameter $\beta_{n_k, n_u}$ approaches zero, the estimator $\boldsymbol{w}_k$ converges to the density ratio $\boldsymbol{w}^*_k$. During federated distillation, we use a threshold $\tau_{\text{client}} > 0$ to distinguish between in-distribution and out-of-distribution samples. If the estimated $\boldsymbol{w}_k(\mathbf{x})$ value of a proxy sample $\mathbf{x}$ is below $\tau_{\text{client}}$, it is considered as an out-of-distribution sample at client $k$. In such cases, the client-side selector does not upload the corresponding local prediction as it could be misleading.

**Server-side selector**. After receiving local predictions from clients, the server averages them to produce the ensemble predictions. For each proxy sample $\mathbf{x}$, the ensemble prediction is denoted as $\boldsymbol{h}^*_{\text{proxy}}(\mathbf{x}) \in \Delta^{C-1}$, and the corresponding one-hot prediction is represented as $\hat{\boldsymbol{h}}^*_{\text{proxy}}(\mathbf{x}) \in V(\Delta^{C-1})$. It is important to note that if the local predictions for a specific proxy sample differ greatly among clients, the resulting ensemble prediction $\boldsymbol{h}^*_{\text{proxy}}(\mathbf{x})$ could be ambiguous with high entropy. This ambiguity could negatively impact knowledge distillation. To address this issue, we developed a server-side selector that measures sample ambiguity by calculating the $\ell_1$ distance between $\boldsymbol{h}^*_{\text{proxy}}(\mathbf{x})$ and $\hat{\boldsymbol{h}}^*_{\text{proxy}}(\mathbf{x})$. The closer this distance is to zero, the less ambiguous the prediction is. In the proposed Selective-FD framework, the server-side selector applies a threshold $\tau_{\text{server}} > 0$ to filter out ambiguous knowledge, where the ensemble predictions with an $\ell_1$ distance greater than $\tau_{\text{server}}$ will not be sent back to the clients for distillation.

---

**Algorithm 1. Selective-FD**

1: Setting the training round $T$ and the client number $K$. Server and clients collect the proxy dataset $\mathcal{D}_{\text{proxy}}$.
2: Clients construct client-side selectors by minimizing (1) and initialize local models.
3: **for** $t$ in 1, ..., $T$
4:   **GenerateEnsemblePredictions**()
5:   **for** Client $k$ in 1, ..., $K$ (in parallel) **do**
6:     Train the local model based on the private dataset $\mathcal{D}_k$.
7:     Utilize proxy samples and ensemble predictions for knowledge distillation.
8:   **end for**
9: **end for**

  **GenerateEnsemblePredictions**():
10: Server randomly selects the indexes of proxy samples and sends them to the clients.
11: **for** Client $k$ in 1, ..., $K$ (in parallel) **do**
12:   Client $k$ computes the predictions on proxy samples, filters out misleading knowledge based on the client-side selector, and uploads the local predictions to the server.
13: **end for**
14: Server aggregates the local predictions, removes ambiguous knowledge based on the server-side selector, and sends the ensemble predictions back to the clients.

## Theoretical insights

In this section, we establish an upper bound for the loss of federated distillation, while also discussing the effectiveness of the proposed selective knowledge sharing mechanism in the context of domain adaptation[51]. To ensure clarity, we begin by providing relevant definitions before delving into the analysis.

**Definition 1**. (Minimum combined loss) The ideal predictor in the hypothesis space $\hat{\mathcal{H}}_k$ achieves the minimum combined loss $\lambda$ over the test and training sets. Two representative $\lambda_k$, $\lambda_{k,\text{proxy}}$ are defined as follows:

$$\lambda_k = \min_{\hat{h}_k \in \hat{\mathcal{H}}_k} \left\{ \mathcal{L}_{\mathcal{D}_{\text{test}}}(\hat{h}_k, \hat{h}^*) + \mathcal{L}_{\mathcal{D}_k}(\hat{h}_k, \hat{h}^*) \right\},$$
$$\lambda_{k,\text{proxy}} = \min_{\hat{h}_k \in \hat{\mathcal{H}}_k} \left\{ \mathcal{L}_{\mathcal{D}_{\text{test}}}(\hat{h}_k, \hat{h}^*) + \mathcal{L}_{\mathcal{D}_{\text{proxy}}}(\hat{h}_k, \hat{h}^*) \right\}. \tag{2}$$

The ideal predictor serves as an indicator of the learning ability of the local model. If the ideal predictor performs poorly, it is unlikely that the locally optimized model, which minimizes the training loss, will generalize well on the test set. On the other hand, when the labeling function $\hat{h}^*$ belongs to the hypothesis space $\hat{\mathcal{H}}_k$, we get the minimum loss as $\lambda_k = \lambda_{k,\text{proxy}} = 0$. The next two definitions aim to introduce a metric for measuring the distance between distributions.

**Definition 2**. (Hypothesis space $\mathcal{G}_k$) For a hypothesis space $\hat{\mathcal{H}}_k$, we define a set of hypotheses $g_k : \mathcal{X} \to \{0,1\}$ as $\mathcal{G}_k$, where $g_k(\mathbf{x}) = \frac{1}{2} \| \hat{h}_k(\mathbf{x}) - \hat{h}_k'(\mathbf{x}) \|_1$ for $\hat{h}_k, \hat{h}_k' \in \hat{\mathcal{H}}_k$.

**Definition 3**. ($\mathcal{G}_k$-distance[52]) Given two distributions $\mathcal{D}$ and $\mathcal{D}'$ over $\mathcal{X}$, let $\mathcal{G}_k = \{g_k : \mathcal{X} \to \{0,1\}\}$ be a hypothesis space, and the $\mathcal{G}_k$-distance between $\mathcal{D}$ and $\mathcal{D}'$ is $d_{\mathcal{G}_k}(\mathcal{D}, \mathcal{D}') = 2 \sup_{g_k \in \mathcal{G}_k} \left| \Pr_{\mathcal{D}}[g_k(\mathbf{x}) = 1] - \Pr_{\mathcal{D}'}[g_k(\mathbf{x}) = 1] \right|$.

With the above preparations, we derive an upper bound of the test loss of the predictor $\hat{h}_k$ at client $k$ following the process of federated distillation.

**Theorem 2**. With probability at least $1 - \delta$, $\delta \in (0, 1)$, we have

$$\mathcal{L}_{\mathcal{D}_{\text{test}}}(\hat{h}_k, \hat{h}^*) \leq \underbrace{\mathcal{L}_{\hat{\mathcal{D}}_k \cup \hat{\mathcal{D}}_{\text{proxy}}}(\hat{h}_k)}_{\text{Empirical risk}} + \underbrace{\sqrt{\left( \frac{2\alpha^2}{m_k} + \frac{2(1-\alpha)^2}{m_{\text{proxy}}} \right) \log \frac{2}{\delta}}}_{\text{Numerical constraint}} \tag{3}$$
$$+ \alpha \left[ \lambda_k + d_{\mathcal{G}_k}(\mathcal{D}_k, \mathcal{D}_{\text{test}}) \right]$$

$$+ (1-\alpha) \left[ \lambda_{k,\text{proxy}} + d_{\mathcal{G}_k}(\mathcal{D}_{\text{proxy}}, \mathcal{D}_{\text{test}}) + \underbrace{p_{\text{proxy}}^{(1)} \mathcal{L}_{\mathcal{D}_{\text{proxy}}^{(1)}}(\hat{h}^*, \hat{h}_{\text{proxy}}^*)}_{\text{Misleading knowledge}} + \underbrace{p_{\text{proxy}}^{(2)} \mathcal{L}_{\mathcal{D}_{\text{proxy}}^{(2)}}(\hat{h}_{\text{proxy}}^*, \hat{h}_{\text{proxy}}^*)}_{\text{Ambiguous knowledge}} \right], \tag{4}$$

where the probabilities $p_{\text{proxy}}^{(1)} = \Pr_{\mathcal{D}_{\text{proxy}}}[\hat{h}^*(\mathbf{x}) \neq \hat{h}_{\text{proxy}}^*(\mathbf{x})]$ and $p_{\text{proxy}}^{(2)} = \Pr_{\mathcal{D}_{\text{proxy}}}[\hat{h}^*(\mathbf{x}) = \hat{h}_{\text{proxy}}^*(\mathbf{x})]$ satisfy $p_{\text{proxy}}^{(1)} + p_{\text{proxy}}^{(2)} = 1$. $\mathcal{D}_{\text{proxy}}^{(1)}$ and $\mathcal{D}_{\text{proxy}}^{(2)}$ represent the distributions of proxy samples satisfying $\hat{h}^*(\mathbf{x}) \neq \hat{h}_{\text{proxy}}^*(\mathbf{x})$ and $\hat{h}^*(\mathbf{x}) = \hat{h}_{\text{proxy}}^*(\mathbf{x})$, respectively.

In (3), the first term on the right-hand side represents the empirical risk over the local and proxy samples, and the second term is a numerical constraint, which indicates that having more proxy samples, whose number is denoted as $m_{\text{proxy}}$, is beneficial to the generalization performance. The last two terms in (4) account for the misleading and ambiguous knowledge in distillation. From Theorem 2, two key implications can be drawn. Firstly, when there is severe data heterogeneity, the resulting high distribution divergence $d_{\mathcal{G}_k}(\mathcal{D}_k, \mathcal{D}_{\text{test}}), d_{\mathcal{G}_k}(\mathcal{D}_{\text{proxy}}, \mathcal{D}_{\text{test}})$ undermines generalization performance. When the proxy distribution is closer to the test set than the local data, i.e., $d_{\mathcal{G}_k}(\mathcal{D}_k, \mathcal{D}_{\text{test}}) \geq d_{\mathcal{G}_k}(\mathcal{D}_{\text{proxy}}, \mathcal{D}_{\text{test}})$, federated distillation can improve performance compared to independent training. Secondly, if the labeling function (i.e., the ensemble prediction) $\hat{h}_{\text{proxy}}^*$ of the proxy samples is highly different from the labeling function $\hat{h}$ of test samples, the error introduced by the misleading and ambiguous

knowledge can be significant, leading to negative knowledge transfer. Our proposed selective knowledge sharing mechanism aims to make the ensemble predictions of unlabeled proxy samples closer to the ground truths. Particularly, a large threshold $\tau_{\text{client}}$ can mitigate the effect of incorrect predictions, while a small threshold $\tau_{\text{server}}$ implies less ambiguous knowledge being used for distillation.

**Remark 1**. Care must be taken when setting the thresholds $\tau_{\text{client}}$ and $\tau_{\text{server}}$, as a $\tau_{\text{client}}$ that is too large or a $\tau_{\text{server}}$ that is too small could filter out too many proxy samples and result in a small $m_{\text{proxy}}$. This would enlarge the second term on the right-hand side of (3). Additionally, the threshold $\tau_{\text{server}}$ effectively balances the losses caused by the misleading and ambiguous knowledge, as indicated by the inequalities $2 - \tau_{\text{server}} \leq \| \hat{h}^*(\mathbf{x}) - \hat{h}_{\text{proxy}}^*(\mathbf{x}) \|_1$ and $\| \hat{h}_{\text{proxy}}^*(\mathbf{x}) - \hat{h}_{\text{proxy}}^*(\mathbf{x}) \|_1 \leq \tau_{\text{server}}$. This property aligns with the empirical results presented in Fig. 6.

**Remark 2**. The proposed mechanism for selectively sharing knowledge and its associated thresholds $\tau_{\text{client}}, \tau_{\text{server}}$ might alter the distributions $\mathcal{D}_{\text{proxy}}, \hat{\mathcal{D}}_{\text{proxy}}$, thus influencing the empirical risk, the minimum combined loss, the $\mathcal{G}_k$-distance, and the probabilities $p_{\text{proxy}}^{(1)}, p_{\text{proxy}}^{(2)}$ in Theorem 2. A more comprehensive and rigorous analysis of these effects is left to our future work.

### Reporting summary

Further information on research design is available in the Nature Portfolio Reporting Summary linked to this article.

### Data availability

The datasets used in this paper are publicly available. The chest X-ray images are from the COVIDx dataset, which is available at https://www.kaggle.com/datasets/andyczhao/covidx-cxr2. Specifically, this dataset consists of NIH Chest X-ray Dataset[53] at https://nihcc.app.box.com/v/ChestXray-NIHCCand RSNA-STR Pneumonia Detection Challenge image datasets[54] at https://www.rsna.org/education/ai-resources-and-training/ai-image-challenge/RSNA-Pneumonia-Detection-Challenge-2018. The benchmark datasets MNIST, Fashion MNIST, and CIFAR-10 are available at http://yann.lecun.com/exdb/mnist/, https://github.com/zalandoresearch/fashion-mnist, and https://www.cs.toronto.edu/-kriz/cifar.html, respectively. The usage of these datasets in this work is permitted under their licenses. Source data are provided with this paper.

### Code availability

Codes[55] for this work are available at https://github.com/shaojiawei07/Selective-FD. All experiments and implementation details are thoroughly described in the Experiments section, Methods section, and Supplementary Information.

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

## Acknowledgements
This work was supported by the Hong Kong Research Grants Council under the Areas of Excellence scheme grant AoE/E-601/22-R (J.Z.) and NSFC/RGC Collaborative Research Scheme grant CRS_HKUST603/22 (J.Z.). Some icons in Fig. 1 are from the website https://icons8.com/. The X-ray icon in Fig. 2 is from Chanut-is-Industries, Flaticon.com.

## Author contributions
J.Z. coordinated and supervised the research project. J.S. conceived the idea of this work, implemented the models for experiments, and analyzed the results. J. S., J.Z., and F.W. contributed to the writing of this manuscript.

## Competing interests
The authors declare no competing interests.
