## [Peer Review File · Nature Communications]

Selective Knowledge Sharing for Privacy-Preserving Federated Distillation without A Good TeacherREVIEWER COMMENTS

Reviewer #1 (Remarks to the Author):

This paper aims to resolve the difficulties of large-size model communication, heterogeneous computation resources, and privacy leakage from parameters in federated learning, by proposing a selective knowledge-sharing mechanism for federated distillation. The proposed method is validated under the non-iid data setting on three natural image datasets and one medical image dataset, it outperforms the compared methods.

Advantages:

- This work aims to study many important problems in FL
- The motivation for using soft/hard labels is clear and helps reduce the communication burden.
- This work presents theoretical insights to demonstrate the impact of the method on training convergence.
- The results for reducing communication burden and handling non-iid data are promising.

Disadvantages:

- The sharing of soft or hard labels raises new concerns about privacy leakage, which may release the class distribution information, it is necessary to assess the risk.
- How to have the proxy data for real applications? The task-related data may be only available at the hospital, the requirement of a proxy dataset requires the hospital to share data, which might be a limitation.
- The setting of strong and weak non-iid is too strict. It is reasonable that some clients may have more than 2 classes, and some clients may have all classes but an imbalanced class distribution.
- The sentence "The architectures of local models are heterogeneous." is a bit misleading. This actually means using different model architectures for different tasks, not assuming for a specific task, the client model architecture varies.
- The architecture is basically the CNN, which does not well support the analysis that other methods suffer high communication overhead when using large-sized FL models.
- The evaluation setting can be further improved. Currently, the comparison only considers one test dataset containing all labels. However, if considering the client data non-iid, then when we test on each client, the testing data should so be non-iid. Under such a local testing setting, the IndepLearn should be very good (IndepLearn shows 10% accuracy on the 10-way classification test dataset, which means

IndepLearn correctly predicts all samples with that specific label). It is necessary to consider the client's local testing dataset and compare all the methods. If we consider the real FL scenario, it is very challenging for the non-iid client to access a global test set.

- The comparison only considers distillation-related methods, the baseline of FedAvg is not included. Also, there are many other methods designed for tackling non-iid data. It would be more convincing to include other related methods in comparison.
- Besides the relative communication cost comparison, what is the difference in terms of the actual number of communication costs (the bytes taken for upload and download).
- For the theoretical insights, the order between different terms is not clear. It seems the empirical risk and the distance dominates the bound. Also, the range of coefficients needs to be clarified. For example, if α is close to 1, then the misleading knowledge and ambiguous knowledge term would have a very limited effect on the error bound. It is necessary to clearly state the order of these terms and the range of the coefficients.
- Better add some experiments regarding the heterogeneous computation resources (e.g., heterogeneous client model architectures) to show the benefits that using FD is agnostic to model client model architectures.

Overall, this paper studies relevant and important topics in FL, for using distillation techniques to solve the issue of communication cost, heterogeneous data, and architectures. The proposed method outperforms all compared KD methods on several datasets under the non-iid setting in this paper. And the proposed method shows a significant reduction in the communication cost. However, the experiment does not fully support the claim, and more experiments are needed. There are some missing aspects like the heterogeneous client model architecture, different evaluation settings, and comparison with other non-distillation-based methods. Moreover, the theoretical insights can be further clarified, some details are not clear. The experimental dataset on real-world application is not large, therefore, its potential impact to the research field is marginal.

Reviewer #2 (Remarks to the Author):

The authors propose a selective knowledge sharing mechanism for federated distillation (an alternative to federated learning). Their method is based on the use of client and server side selectors to filter out misleading knowledge and therefore improve on existing baseline methods.

The authors clearly explain their solution and why it should conceptually yield better results than existing FD approaches. They also experimentally compare their approach with multiple existing solutions on 3 image datasets, including one on COVID-19 pneumonia.

They also clearly highlight the flaws of existing methods, including those of federated learning (FL).

On the comparison with FL, two points raised by the authors should be more clearly justified/mentioned. The first one is that FD and FL consider two different settings and cannot be directly compared. The second is that it is not clear whereas FD is in general more private than FL, and how this difference in terms of privacy could be actually quantified. The possible information leakage from the sharing of proxy samples and labels is difficult to assess. For a fair discussion, the authors should at least mention the existence (or not) of studies on this leakage, similarly to what they did for FL.

For a more comprehensive and fair comparison with existing FD solutions, that seem to not be adapted for the non-iid. setting, the authors should also compare their approach with other solutions in the iid. scenario. If doable, to strengthen their contribution, the authors should compare their solution against another solution that is designed to handle the non-iid. setting.^[1] The comparison in “Communication Overhead”, should be more extensively described, especially on how the authors ensure a fair comparison while using different paradigms.

The authors should introduce the thresholds and their purpose in the beginning of the “Threshold Analysis”. The thresholds are only introduced in Methods afterwards.

The authors propose a conceptually simple solution for FD and show that it outperforms multiple existing solutions on multiple datasets. The paper is clearly structured and easy to read. However, the authors should clearly highlight and explain their contribution by providing a fair and comprehensive comparison, and clearly explain how their solution advances the state of the art.

Authors' Response to Reviews of

Selective Knowledge Sharing for Privacy-Preserving Federated Distillation without A Good Teacher

Jiawei Shao, Fangzhao Wu, and Jun Zhang
Nature Communications,

RC: *Reviewers' Comment*, AR: Authors' Response, □ Manuscript Text

Overview

We are very grateful to the reviewers for their constructive comments. We have carefully prepared the response letter and revised the manuscript based on the valuable feedback. To address the concern regarding the impact of our work, we have provided a **general response** to highlight our motivations and contributions. Besides, the changes made in the revision are highlighted in blue, and the major ones are summarized below.

- To address Reviewer 1's concern about the theoretical insights, we have empirically evaluated the influence of parameter α in Theorem 2 on model performance.
- As suggested by Reviewer 1, we have reported the download and upload communication costs in the manuscript.
- To address the privacy concern raised by Reviewer 2, we have compared the privacy leakage of the proposed method with FedAvg under model inversion attacks.
- As suggested by Reviewer 2, we have evaluated the performance of the proposed method and baselines in the IID setting.

All the comments raised by the reviewers have been considered in the revision of our manuscript. The following response letter addresses all the comments in detail.

Thank you very much for dedicating your time and effort towards enhancing the quality of our paper. We look forward to hearing from you again.

General Response: The Impact of Our Work.

Comments

- **Reviewer 1:** This paper studies relevant and important topics in FL, for using distillation techniques to solve the issue of communication cost, heterogeneous data, and architectures. The proposed method outperforms all compared KD methods on several datasets under the non-IID setting in this paper. And the proposed method shows a significant reduction in the communication cost. However, the experiment does not fully support the claim, and more experiments are needed. There are some missing aspects like the heterogeneous client model architecture, different evaluation settings, and comparison with other non-distillation-based methods. Moreover, the theoretical insights can be further clarified, some details are not clear. The experimental dataset on the real-world application is not large, therefore, its potential impact on the research field is marginal.
- **Reviewer 2:** The authors propose a conceptually simple solution for FD and show that it outperforms multiple existing solutions on multiple datasets. The paper is clearly structured and easy to read. However, the authors should clearly highlight and explain their contribution by providing a fair and comprehensive comparison, and clearly explain how their solution advances the state of the art.

Authors' response

Thanks for these comments. Regarding the impact of our work, the response is as follows:

Background: We are currently experiencing an extraordinary surge in global data traffic, a trend that is being accelerated by the increasing popularity of various computing devices. A forecast by International Data Corporation estimates that there will be 41.6 billion IoT devices in 2025, capable of generating 79.4 zettabytes of data [Ref-1]. The unprecedented amount of data, together with the fast development of artificial intelligence, e.g., neural networks and deep learning (DL) models, has driven the development of innovative data-driven solutions in a range of fields, leading to profound economic and societal impacts. However, with the increasing privacy awareness, the traditional centralized approach of training DL models, i.e., collecting distributed data to a powerful server for model training, is facing strong challenges. Sensitive data such as credit card numbers, medical records, and location-based services, can be used for targeted advertising and personalized recommendations, posing potential privacy risks. The emerging legal restrictions, such as the General Data Protection Regulation (GDPR), California Consumer Privacy Act (CCPA), and Health Insurance Portability and Accountability Act (HIPAA), make data aggregation practices less feasible.

Federated learning: Federated learning (FL) is a promising solution to the new reality, effectively utilizing distributed data while preserving privacy. Specifically, multiple data-owning clients collaborate to train DL models. This is accomplished by updating models locally using private data, and then aggregating these updates globally. These two steps iterate many times until convergence. One main advantage of FL is that private data does not need to be moved or centralized in one location for training. However, sharing model parameters in FL brings new challenges and inconveniences.

- **Privacy leakage:** Although FL does not require data sharing, recent works have demonstrated that FL may not always provide sufficient privacy guarantees, as communicating models throughout the training process can nonetheless reveal sensitive information. As discussed in [Ref-2], DL models

	Federated distillation (FD)	Standard federated learning (FL)
Representative methods	FedMD, FedED	FedAvg, FedProx
What to share	Knowledge	Model parameters
Higher ability to mitigate the non-IID issue		✓
Lower communication overhead	✓	
Lower privacy leakage	✓	
Better adaptation to heterogeneous models	✓	

Table 1: Comparison between federated distillation (FD) and standard federated learning (FL).

represent a form of memory mechanism, with compressed representations of the training data stored within their weights. It is therefore possible for a semi-honest server in FL to reconstruct parts of the training data from the model parameters.

- **Communication bottleneck:** During the training process, the frequent transfer of models between clients and the server generates substantial communication costs. This becomes particularly challenging when dealing with large language models, such as GPT and LLaMA. These models comprise billions of parameters, thereby bringing the curse of prohibitive communication overhead.
- **Difficult to support heterogeneous clients:** The server in FL averages local models for aggregation, necessitating that all local models share the same architecture. However, clients in practice may have varying power supplies, memory capacities, and computational resources. By initializing the same model across these diverse clients, some can update local models quickly while others may take longer. Stragglers in the FL settings could significantly prolong the training time.

Federated distillation: Federated distillation (FD), built upon knowledge distillation—an effective technique for transferring knowledge from teacher models to student models—**naturally tackles the above difficulties by sharing knowledge among clients instead of model parameters.** However, one challenge of FD is the absence of a high-quality teacher model. At the beginning of the training process, all the local models are randomly initialized, and the ensemble of local predictors is used for knowledge distillation. Therefore, the federated training process is sensitive to the state of local models. If the local models are underfitting, the local predictions could be highly misleading. Besides, the non-IID data distributions across clients exacerbate this issue, since the local models cannot output accurate predictions on the samples that are outside their local distributions. Table 1 summarizes a comparison between FD and standard FL settings.

Our contributions and impacts: Our work improves the performance of knowledge distillation in FD when the teacher model is absent. We proposed a *selective knowledge sharing* mechanism in federated distillation (named *Selective-FD*) to filter out misleading knowledge during the training process. Extensive experimental results, backed by theoretical analysis, show that Selective-FD excels in handling non-IID data and significantly improves test accuracy compared to baselines.

In comparison with standard FL settings, FD improves the privacy guarantees, reduces the communication overhead, and has the capability to support heterogeneous clients. Previously, a significant challenge impeding the practical implementation of FD is its noticeably inferior performance compared to FL. In this study, we tackle this issue by introducing a selective knowledge sharing mechanism. Our method substantially enhances

test accuracy, closing the performance gap with the FL method. We foresee that the proposed method will be a valuable tool for training models in an efficient and privacy-preserving manner. It is promising especially for scenarios with large model sizes, e.g., the emerging large language models (LLMs). Thus, we envision this study will lead to substantial impacts.

Revisions in the manuscript: As suggested by the reviewers, we have provided additional details about the experimental setups and conducted supplementary experiments for comprehensive comparisons. Further elaboration on the revisions can be found in the subsequent responses.

[Ref-1] Wang, William Yu Chung, and Yichuan Wang. Analytics in the era of big data: The digital transformations and value creation in industrial marketing. *Industrial Marketing Management* 86, 12-15 (2020).

[Ref-2] Kaissis, Georgios A., et al. Secure, privacy-preserving and federated machine learning in medical imaging. *Nature Machine Intelligence* 2.6, 305-311 (2020).

Reviewer 1

RC: *This paper aims to resolve the difficulties of large-size model communication, heterogeneous computation resources, and privacy leakage from parameters in federated learning, by proposing a selective knowledge-sharing mechanism for federated distillation. The proposed method is validated under the non-IID data setting on three natural image datasets and one medical image dataset, it outperforms the compared methods.*

- *This work aims to study many important problems in FL.*
- *The motivation for using soft/hard labels is clear and helps reduce the communication burden.*
- *This work presents theoretical insights to demonstrate the impact of the method on training convergence.*
- *The results for reducing communication burden and handling non-IID data are promising.*

AR: Thank you for acknowledging our contributions. We have looked into your comments carefully and revised the manuscript accordingly. Our point-to-point responses to your comments are given below.

Comment 1.1

RC: *The sharing of soft or hard labels raises new concerns about privacy leakage, which may release the class distribution information, it is necessary to assess the risk.*

AR: We agree with the reviewer that federated distillation may enable a semi-honest server to infer the clients' local class distribution based on the shared hard or soft labels. An effective way to solve this problem is secure aggregation [Ref-1.1-1], which privately aggregates the shared information from clients without revealing the individual information. In this scenario, the semi-honest server cannot access the shared labels from a specific client and thus lacks the ability to infer the local class distribution. We have added a related discussion in the manuscript.

[Ref-1.1-1] Bonawitz, Keith, et al. Practical secure aggregation for privacy-preserving machine learning. In *SIGSAC Conference on Computer and Communications Security* (2017).

Comment 1.2

RC: *How to have the proxy data for real applications? The task-related data may be only available at the hospital, the requirement of a proxy dataset requires the hospital to share data, which might be a limitation.*

AR: Thank you for this question. In practice, we can collect the proxy data from publicly available datasets that are similar to the target application. The data platforms such as OpenDataLab, Paperswithcode, and HuggingFace provide extensive resources to accelerate the reuse of public datasets. Take the pneumonia detection task as an example. The proxy X-ray images can be produced from the public chest computerized tomography (CT) scan dataset. Another alternative method to tackle this problem is generating synthetic data for knowledge distillation [Ref-1.2-1, Ref-1.2-2]. This involves training generators based on the local datasets and sharing generated samples among clients as proxy data.

In the revision, we have provided more discussion on the proxy dataset in the Supplementary Information.

Figure 1: MNIST classification accuracy in different non-IID settings. When β is set to a smaller value, the data distribution is more non-IID. The knowledge is transferred via (a) hard labels and (b) soft labels.

[Ref-1.2-1] Zhu, Zhuangdi, et al. Data-free knowledge distillation for heterogeneous federated learning. In *International Conference on Machine Learning* (2021).

[Ref-1.2-2] Lin, Tao, et al. Ensemble distillation for robust model fusion in federated learning. *Advances in Neural Information Processing Systems* (2020).

Comment 1.3

RC: *The setting of strong and weak non-IID is too strict. It is reasonable that some clients may have more than 2 classes, and some clients may have all classes but an imbalanced class distribution.*

AR: Thank you for this suggestion, following which we have added new experiments with more general non-IID settings. Following [Ref-1.3-1, Ref-1.3-2], we simulate the non-IID distribution based on the Dirichlet distribution $\text{Dir}_K(\beta)$, where K represents the number of clients. Here $\beta > 0$ is a concentration parameter. When β is set to a smaller value, the data distribution is more non-IID. Specifically, we sample a vector $p_n \sim \text{Dir}_K(\beta)$ and allocate a $p_{n,k}$ proportion of the instances of class n to client k . We conduct a performance comparison of the proposed Selective-FD method with two representative baselines, namely FedMD and IndepLearn, across various non-IID settings of the MNIST dataset. As shown in Fig. 1, the accuracy of IndepLearn degrades with the decreasing of parameter β . This is because the data distribution becomes increasingly non-IID. The FedMD method demonstrates the ability to achieve good performance in weak non-IID scenarios ($10^{-1} < \beta$). However, it experiences performance degradation in strong non-IID settings ($\beta < 10^{-1}$). In contrast, our Selective-FD method consistently maintains satisfactory performance, even when the parameter β decreases to 10^{-3} . Specifically, the accuracy gain of our method is more significant when using hard labels for knowledge distillation. These empirical results are consistent with Table 8 (Table 1 in the manuscript).

In the revision, we have added this experiment in the Supplementary Information.

[Ref-1.3-1] Yurochkin, Mikhail, et al. Bayesian nonparametric federated learning of neural networks. In *International Conference on Machine Learning* (2019).

[Ref-1.3-2] Li, Qinbin, et al. Federated learning on non-iid data silos: An experimental study. In *International Conference on Data Engineering* (2022).

Comment 1.4

RC: *The sentence “The architectures of local models are heterogeneous.” is a bit misleading. This actually means using different model architectures for different tasks, not assuming for a specific task, the client model architecture varies.*

AR: Sorry for the confusion. We use the term *local models* to denote the clients’ models for a specific task in FL. The federated distillation framework enables collaborative training for heterogeneous models, allowing each client to have a unique model structure. We have revised the manuscript to improve the clarity.

Comment 1.5

RC: *The architecture is basically the CNN, which does not well support the analysis that other methods suffer high communication overhead when using large-sized FL models.*

AR: Thank you for this comment. The authors would like to bring to your attention that the high communication overhead is a main bottleneck in federated learning, which has been widely discussed in the literature [Ref-1.5-1, Ref-1.5-2, Ref-1.5-3]. As large-scale deep learning models with the CNN architecture, e.g., VGG [Ref-1.5-4] and ResNet [Ref-1.5-5], contain millions of parameters, the iterative process of exchanging model updates in FL leads to high training latency. Table 2 and Fig. 2 (Table 2 and Fig. 7 in the manuscript) demonstrate that FedAvg incurs much higher communication costs than our Selective-FD method.

[Ref-1.5-1] Lim, Wei Yang Bryan, et al. Federated learning in mobile edge networks: A comprehensive survey. *IEEE Communications Surveys & Tutorials* (2020).

[Ref-1.5-2] Li, Tian, et al. Federated learning: Challenges, methods, and future directions. *IEEE signal processing magazine* (2020).

[Ref-1.5-3] Niknam, Solmaz, et al. Federated learning for wireless communications: Motivation, opportunities, and challenges. *IEEE Communications Magazine* (2020).

[Ref-1.5-4] Simonyan, K., and A. Zisserman. Very deep convolutional networks for large-scale image recognition. In *International Conference on Learning Representations* (2015).

[Ref-1.5-5] He, Kaiming, et al. Deep residual learning for image recognition. In *Proceedings of the IEEE Conference on Computer Vision and Pattern Recognition* (2016).

Comment 1.6

RC: *The evaluation setting can be further improved. Currently, the comparison only considers one test dataset containing all labels. However, if considering the client data non-IID, then when we test on each client, the testing data should so be non-IID Under such a local testing setting, the IndepLearn should be very good (IndepLearn shows 10% accuracy on the 10-way classification test dataset, which means IndepLearn correctly predicts all samples with that specific label). It is necessary to consider the client’s local testing dataset and compare all the methods. If we consider the real FL scenario, it is very challenging for the non-IID client to access a global test set.*

AR: We greatly appreciate the reviewer’s insightful comment regarding the evaluation in non-IID settings. However,

Figure 2: Test accuracy and communication cost as functions of the communication round. Our Selective-FD method has a comparable accuracy as FedAvg in the MNIST and CIFAR-10 datasets but is inferior in Fashion MNIST. However, Selective-FD achieves a significant reduction in the communication overhead, which is because the cost of model sharing in FedAvg is much higher than knowledge sharing in our method.

we respectfully disagree with the suggestion that test data for each client’s model should be different.

Consider the strong non-IID setting, where each client has only one class of data. If the test data of each client follows the same distribution as the respective training set, IndepLearn achieves the best test accuracy of 100% by overfitting the local training set. However, this perfect performance is misleading, as it lacks generalization ability and cannot perform accurately against unseen data. In practice, if hospitals employ overfitting models to detect pneumonia, the consequences could be severe, potentially leading to a high rate of misdiagnosis.

Comment 1.7

RC: *The comparison only considers distillation-related methods, and the baseline of FedAvg is not included. Also, there are many other methods designed for tackling non-IID data. It would be more convincing to include other related methods in comparison.*

AR: Thank you for the suggestions. We would like to kindly draw your attention to the fact that the test accuracy of FedAvg has already been reported in Fig. 2 (Fig. 7 in the manuscript). It is observed that our Selective-FD method achieves comparable or inferior accuracy to FedAvg. But a significant advantage of Selective-FD is its huge reduction in communication overhead.

We agree with the reviewer that there are many federated learning (FL) methods designed for tackling the non-IID issue. However, it is important to note that the proposed Selective-FD is fundamentally a knowledge distillation (FD) approach. Comparing the performance of FD with the standard FL framework [Ref-1.7-1, Ref-1.7-2] is unfair. Table 1 summarizes the difference between FD and FL.

- The standard FL methods exchange model parameters among the server and clients, while the clients in FD share knowledge for model distillation. As the shared model parameters inherently contain more information than the shared knowledge in FD, it will be easier for FL to tackle the non-IID issue. For example, FedProx [Ref-1.7-2] adds a proximal term in the local objective function to minimize the difference between the local model and the global model. This effectively mitigates the model

Communication overhead	Selective-FD			FedAvg	
	Collect proxy samples	Download	Upload	Download	Upload
MNIST	4.7 MB	21.5 KB	20.5 KB	87.4 KB	87.4 KB
Fashion MNIST	4.7 MB	21.5 KB	20.5 KB	7.46 MB	7.46 MB
CIFAR-10	30.7 MB	1.3 KB	1.3 KB	45.1 MB	45.1 MB

Table 2: Communication overhead of each client. Note that collecting proxy samples only incurs a one-time cost, while downloading and uploading are needed for each iteration.

divergence caused by the non-IID data.

- As shown in Fig. 2, FD may not consistently match the performance of FL, but it is more communication-efficient since the size of knowledge is much smaller than the size of the local model. Besides, unlike FL, FD is free from white-box privacy attacks since the local model remains inaccessible to other participants [Ref-1.7-3]. Furthermore, FD is agnostic to the local model structure and thus can adapt to heterogeneous clients.

[Ref-1.7-1] McMahan, Brendan, et al. Communication-efficient learning of deep networks from decentralized data. *Artificial intelligence and statistics* (2017).

[Ref-1.7-2] Li, Tian, et al. Federated optimization in heterogeneous networks. In *Proceedings of Machine learning and systems* (2020).

[Ref-1.7-3] Mothukuri, Viraaji, et al. A survey on security and privacy of federated learning. *Future Generation Computer Systems* (2021).

Comment 1.8

RC: *Besides the relative communication cost comparison, what is the difference in terms of the actual number of communication costs (the bytes taken for upload and download)?*

AR: Thank you for this comment. Table 2 (Table 2 in the manuscript) summarizes the communication overhead in federated training. Our Selective-FD method, while introducing a one-time communication cost for collecting proxy samples, significantly reduces upload and download costs per communication round compared with FedAvg.

In the revision, we have added this result in the “Communication Overhead” subsection.

Comment 1.9

RC: *For the theoretical insights, the order between different terms is not clear. It seems the empirical risk and the distance dominates the bound. Also, the range of coefficients needs to be clarified. For example, if α is close to 1, then the misleading knowledge and ambiguous knowledge term would have a very limited effect on the error bound. It is necessary to clearly state the order of these terms and the range of the coefficients.*

Figure 3: Test error rate as a function of α .

AR: Thank you for this comment. Recap the upper bound of the test loss in Theorem 2:

$$\begin{aligned}
\mathcal{L}_{\mathcal{D}_{\text{test}}}(\hat{\mathbf{h}}_k, \hat{\mathbf{h}}^*) &\leq \underbrace{\mathcal{L}_{\hat{\mathcal{D}}_k \cup \hat{\mathcal{D}}_{\text{proxy}}}(\hat{\mathbf{h}}_k)}_{\text{Empirical risk}} + \underbrace{\sqrt{\left(\frac{2\alpha^2}{m_k} + \frac{2(1-\alpha)^2}{m_{\text{proxy}}}\right) \log \frac{2}{\delta}}}_{\text{Numerical constraint}} + \alpha [\lambda_k + d_{\mathcal{G}_k}(\mathcal{D}_k, \mathcal{D}_{\text{test}})] \\
&+ (1-\alpha) \left[\lambda_{k,\text{proxy}} + d_{\mathcal{G}_k}(\mathcal{D}_{\text{proxy}}, \mathcal{D}_{\text{test}}) + \underbrace{p_{\text{proxy}}^{(1)} \mathcal{L}_{\mathcal{D}_{\text{proxy}}^{(1)}}(\hat{\mathbf{h}}^*, \mathbf{h}_{\text{proxy}}^*)}_{\text{Misleading knowledge}} + \underbrace{p_{\text{proxy}}^{(2)} \mathcal{L}_{\mathcal{D}_{\text{proxy}}^{(2)}}(\hat{\mathbf{h}}_{\text{proxy}}^*, \mathbf{h}_{\text{proxy}}^*)}_{\text{Ambiguous knowledge}} \right], \tag{1}
\end{aligned}$$

where the empirical risk $\mathcal{L}_{\hat{\mathcal{D}}_k \cup \hat{\mathcal{D}}_{\text{proxy}}}(\hat{\mathbf{h}}_k) := \alpha \mathcal{L}_{\hat{\mathcal{D}}_k}(\hat{\mathbf{h}}_k, \hat{\mathbf{h}}^*) + (1-\alpha) \mathcal{L}_{\hat{\mathcal{D}}_{\text{proxy}}}(\hat{\mathbf{h}}_k, \mathbf{h}_{\text{proxy}}^*)$ and

$$\lambda_k = \min_{\hat{\mathbf{h}}_k \in \hat{\mathcal{H}}_k} \left\{ \mathcal{L}_{\mathcal{D}_{\text{test}}}(\hat{\mathbf{h}}_k, \hat{\mathbf{h}}^*) + \mathcal{L}_{\mathcal{D}_k}(\hat{\mathbf{h}}_k, \hat{\mathbf{h}}^*) \right\}, \quad \lambda_{k,\text{proxy}} = \min_{\hat{\mathbf{h}}_k \in \hat{\mathcal{H}}_k} \left\{ \mathcal{L}_{\mathcal{D}_{\text{test}}}(\hat{\mathbf{h}}_k, \hat{\mathbf{h}}^*) + \mathcal{L}_{\mathcal{D}_{\text{proxy}}}(\hat{\mathbf{h}}_k, \hat{\mathbf{h}}^*) \right\}. \tag{2}$$

Prior to investigating the effect of coefficient α on the error bound, we first identify the terms in (1) that are negligible. Consider that the deep learning model at client k has enough parameters such that its hypothesis space $\hat{\mathcal{H}}_k$ contains the ground truth labeling function $\hat{\mathbf{h}}^*$. In this case, both λ_k and $\lambda_{k,\text{proxy}}$ in (2) hold a value of zero. Besides, the numerical constraint tends to be zero given sufficient training samples. Furthermore, the proxy dataset $\mathcal{D}_{\text{test}}$ for knowledge distillation is expected to be less heterogeneous compared with the local heterogeneous dataset \mathcal{D}_k at client k . Thus, the distance $\mathcal{G}_k(\mathcal{D}_{\text{proxy}}, \mathcal{D}_{\text{test}})$ is much smaller than $\mathcal{G}_k(\mathcal{D}_k, \mathcal{D}_{\text{test}})$. If the proxy dataset follows the same distribution as the test set, then the $\mathcal{G}_k(\mathcal{D}_{\text{proxy}}, \mathcal{D}_{\text{test}})$ distance equals to zero.

Now it is clear that when α approaches 1, the empirical risk and the distance $\mathcal{G}_k(\mathcal{D}_k, \mathcal{D}_{\text{test}})$ dominate the error bound. When α is close to 0, the empirical risk, misleading knowledge, and ambiguous knowledge become the dominant factors. To support this analysis, we conduct an ablation study on MNIST under a weak non-IID

Client 1	Client 2	Client 3	Client 4	Client 5	Client 6	Client 7	Client 8	Client 9	Client 10
ResNet*-18	ResNet*-18	ResNet*-18	ResNet*-18	ResNet*-34	ResNet*-34	ResNet*-34	ResNet*-34	ResNet*-50	ResNet*-50
Linear(128)	Linear(128)	Linear(256)	Linear(256)	Linear(128)	Linear(128)	Linear(256)	Linear(256)	Linear(256)	Linear(256)
ReLU	ReLU	ReLU	ReLU	ReLU	ReLU	ReLU	ReLU	ReLU	ReLU
Linear(64)	Linear(64)	Linear(10)	Linear(10)	Linear(64)	Linear(64)	Linear(10)	Linear(10)	Linear(10)	Linear(10)
ReLU	ReLU			ReLU	ReLU				
Linear(10)	Linear(10)			Linear(10)	Linear(10)				

Table 3: The architectures of local models on the CIFAR-10 dataset. ResNet* represents the layers of ResNet before the fully connected layer. The fully-connected layer with output dimension o is defined as linear(o), and ReLU represents the rectified linear unit function.

setting, comparing the classification error across various α values. Our method utilizes the proposed selection mechanism as a pseudo-labeling function of the proxy dataset $\mathbf{h}_{\text{proxy}}^*$. The local client k trains its local model by minimizing the empirical risk $\alpha \mathcal{L}_{\mathcal{D}_k}(\hat{\mathbf{h}}_k, \hat{\mathbf{h}}^*) + (1 - \alpha) \mathcal{L}_{\mathcal{D}_{\text{proxy}}}(\hat{\mathbf{h}}_k, \mathbf{h}_{\text{proxy}}^*)$. To better assess the negative impact of misleading and ambiguous knowledge, we consider a baseline where the proxy dataset has ground truth labels $\hat{\mathbf{h}}^*(\mathbf{x})$. The client minimizes the combined loss $\alpha \mathcal{L}_{\mathcal{D}_k}(\hat{\mathbf{h}}_k, \hat{\mathbf{h}}^*) + (1 - \alpha) \mathcal{L}_{\mathcal{D}_{\text{proxy}}}(\hat{\mathbf{h}}_k, \hat{\mathbf{h}}^*)$ to train the local model.

As shown in Fig. 3, the test error rate of the baseline decreases monotonously as the α value decreases. This is because a small α reduces the negative influence of the local heterogeneous dataset \mathcal{D}_k on the training process. In contrast, the error rate of our proposed method first decreases but then increases when α approaches 0. These results can primarily be attributed to misleading and ambiguous knowledge, which degrades the training performance, particularly when α is close to 0.

It is important to note that calculating the terms $d_{\mathcal{G}_k}(\mathcal{D}_k, \mathcal{D}_{\text{test}})$, $d_{\mathcal{G}_k}(\mathcal{D}_{\text{proxy}}, \mathcal{D}_{\text{test}})$, $p_{\text{proxy}}^{(1)} \mathcal{L}_{\mathcal{D}_{\text{proxy}}^{(1)}}(\hat{\mathbf{h}}^*, \mathbf{h}_{\text{proxy}}^*)$, $p_{\text{proxy}}^{(2)} \mathcal{L}_{\mathcal{D}_{\text{proxy}}^{(2)}}(\hat{\mathbf{h}}_{\text{proxy}}^*, \mathbf{h}_{\text{proxy}}^*)$ in (1) is difficult, since the data distributions \mathcal{D}_k , $\mathcal{D}_{\text{test}}$, $\mathcal{D}_{\text{proxy}}$ are generally unknown in practice. The optimal value of α must be discovered via trial and error. Fortunately, Fig. 3 shows that the test error rate is not significantly affected by α within the range of $[0.2, 0.8]$.

In the revised version, additional discussion about theoretical insights can be found in the Supplementary Information.

Comment 1.10

RC: *Better add some experiments regarding the heterogeneous computation resources (e.g., heterogeneous client model architectures) to show the benefits that using FD is agnostic to model client model architectures.*

AR: Thank you for the comment. Many works have discussed the advantages of the model-agnostic training framework in FL [Ref-1.10-1, Ref-1.10-2, Ref-1.10-3]. Following the reviewer’s suggestion, we provide an example based on the CIFAR-10 classification task. Consider that clients 1 to 4 utilize Jetson AGX Orin modules to train the models. Clients 5 to 8 and clients 9 to 10 own GPU servers equipped with the NVIDIA GeForce GTX 1080 Ti graphics card and NVIDIA GeForce RTX 2080 Ti graphics cards, respectively. Selective-FD allows clients to use different models for local training. A detailed summary of these model structures can be found in Table 5. On the other hand, FedAvg initializes the global model as ResNet-50 in this experiment. The computation latency per round is shown in Fig. 4. Compared with FedAvg, clients 1 to 4 in Selective-FD take half the computation time in each federated training round. This demonstrates the effectiveness of the model-agnostic FL method to reduce training complexity.

Figure 4: Computation latency per communication round in federated training on CIFAR-10.

In our revised version, we have incorporated this example into the Supplementary Information.

[Ref-1.10-1] Fallah, Alireza, et al. Personalized federated learning with theoretical guarantees: A model-agnostic meta-learning approach. *Advances in Neural Information Processing Systems* (2020).

[Ref-1.10-2] Zhu, Zhuangdi, et al. Data-free knowledge distillation for heterogeneous federated learning. In *International Conference on Machine Learning* (2021).

[Ref-1.10-3] Lin, Tao, et al. Ensemble distillation for robust model fusion in federated learning. *Advances in Neural Information Processing Systems* (2020).

Reviewer 2

RC: *The authors propose a selective knowledge sharing mechanism for federated distillation (an alternative to federated learning). Their method is based on the use of client and server side selectors to filter out misleading knowledge and therefore improve on existing baseline methods. The authors clearly explain their solution and why it should conceptually yield better results than existing FD approaches. They also experimentally compare their approach with multiple existing solutions on 3 image datasets, including one on COVID-19 pneumonia. They also clearly highlight the flaws of existing methods, including those of federated learning (FL). The authors propose a conceptually simple solution for FD and show that it outperforms multiple existing solutions on multiple datasets. The paper is clearly structured and easy to read. However, the authors should clearly highlight and explain their contribution by providing a fair and comprehensive comparison, and clearly explain how their solution advances the state of the art.*

AR: We thank the reviewer for the positive feedback. We have conducted more experiments to provide comprehensive comparisons with baselines. Our detailed responses to your comments are given below.

Comment 2.1

RC: *In the comparison with FL, two points raised by the authors should be more clearly justified/mentioned. The first one is that FD and FL consider two different settings and cannot be directly compared. The second is that it is not clear whether FD is in general more private than FL, and how this difference in terms of privacy could be actually quantified. The possible information leakage from the sharing of proxy samples and labels is difficult to assess. For a fair discussion, the authors should at least mention the existence (or not) of studies on this leakage, similarly to what they did for FL.*

RC: *The comparison in “Communication Overhead”, should be more extensively described, especially on how the authors ensure a fair comparison while using different paradigms.*

AR: Thank you for these comments. The settings of FD and FL are as follows:

- Table 4 and Table 5 summarize the heterogeneous local models in FD. Besides, Table 6 shows the global model initialized in FL. The convolutional layer with output channel o , kernel size k , and padding p is denoted as $\text{Conv}(o, k, p)$. The fully-connected layer with output dimension o is defined as $\text{linear}(o)$, and the max-pooling layer with kernel size k is denoted as $\text{MaxPool}(k)$. ReLU represents the rectified linear unit function.
- In each communication round, every client in FD performs 1-step and 10-step SGD training on the local dataset and proxy dataset, respectively. To ensure a fair comparison, the clients in FL perform 11-step local SGD training on the local data.

More details about the experiments have been deferred to Supplementary Information.

Next, we discuss the privacy leakage of FD and FL. The FL framework requires clients to iteratively upload the model updates to the server, which enables a semi-honest server to perform white-box model inversion attacks to reconstruct the private samples from the model parameters [Ref-2.1-1]. On the contrary, the FD methods are free from such privacy attacks since the server and other clients cannot access the clients’ models. However, FD is still vulnerable to black-box attacks, where the attacker can infer local samples by querying the clients’ models [Ref-2.1-2]. To quantitatively evaluate the privacy risk, we perform model inversion attacks on the proposed Selective-FD and FedAvg. In particular, we employ [Ref-2.1-1] and [Ref-2.1-2] to

Client 1	Client 2	Client 3	Client 4	Client 5	Client 6	Client 7	Client 8	Client 9	Client 10
Conv(10, 5, 0)	Conv(10, 5, 0)	Conv(10, 3, 1)	Conv(10, 3, 1)	Conv(10, 5, 0)	Conv(10, 5, 0)	Linear(1024)	Linear(1024)	Linear(1024)	Linear(1024)
ReLU	ReLU	ReLU	ReLU	ReLU	ReLU	ReLU	ReLU	ReLU	ReLU
MaxPool(2)	MaxPool(2)	MaxPool(2)	MaxPool(2)	MaxPool(2)	MaxPool(2)	Linear(512)	Linear(512)	Linear(1024)	Linear(1024)
Conv(20, 5, 0)	Conv(20, 5, 0)	Conv(20, 3, 1)	Conv(20, 3, 1)	Conv(20, 3, 1)	Conv(20, 3, 1)	ReLU	ReLU	ReLU	ReLU
ReLU	ReLU	ReLU	ReLU	ReLU	ReLU	Linear(256)	Linear(256)	Linear(10)	Linear(10)
MaxPool(2)	MaxPool(2)	MaxPool(2)	MaxPool(2)	MaxPool(2)	MaxPool(2)	ReLU	ReLU	ReLU	ReLU
Linear(50)	Linear(50)	Linear(128)	Linear(128)	Linear(64)	Linear(64)	Linear(10)	Linear(10)		
ReLU	ReLU	ReLU	ReLU	ReLU	ReLU				
Linear(10)	Linear(10)	Linear(10)	Linear(10)	Linear(10)	Linear(10)				

Table 4: The architectures of local models on the MNIST and Fashion MNIST datasets.

Client 1	Client 2	Client 3	Client 4	Client 5	Client 6	Client 7	Client 8	Client 9	Client 10
ResNet*-18	ResNet*-18	ResNet*-18	ResNet*-18	ResNet*-34	ResNet*-34	ResNet*-34	ResNet*-34	ResNet*-50	ResNet*-50
Linear(128)	Linear(128)	Linear(256)	Linear(256)	Linear(128)	Linear(128)	Linear(256)	Linear(256)	Linear(256)	Linear(256)
ReLU	ReLU	ReLU	ReLU	ReLU	ReLU	ReLU	ReLU	ReLU	ReLU
Linear(64)	Linear(64)	Linear(10)	Linear(10)	Linear(64)	Linear(64)	Linear(10)	Linear(10)	Linear(10)	Linear(10)
ReLU	ReLU			ReLU	ReLU				
Linear(10)	Linear(10)			Linear(10)	Linear(10)				

Table 5: The architectures of local models on the CIFAR-10 dataset. ResNet* represents the layers of ResNet before the fully connected layer.

perform white-box attacks and black-box attacks, respectively. The experimental results are shown in Fig. 5 (Fig. 8 in the manuscript). It is observed that the quality of reconstructed images from FedAvg is better than that from Selective-FD. This demonstrates that sharing model parameters lead to higher privacy leakage than sharing knowledge. Besides, sharing hard labels in Selective-FD exposes less private information than sharing soft labels.

Third, to better compare the communication overhead of Selective-FD with FedAvg, we show the upload and download costs per communication round in Table 7 (Table 2 in the manuscript). Our Selective-FD method, while introducing a one-time communication cost for collecting proxy samples, significantly reduces upload and download costs compared with FedAvg.

[Ref-2.1-1] Zhang, Yuheng, et al. The secret revealer: Generative model-inversion attacks against deep neural networks. In *Proceedings of the Conference on Computer Vision and Pattern Recognition* (2020).

[Ref-2.1-2] Zhang, Jie, et al. IDEAL: Query-Efficient Data-Free Learning from Black-Box Models. In *International Conference on Learning Representations*. (2022).

Comment 2.2

RC: *For a more comprehensive and fair comparison with existing FD solutions, that seem to not be adapted for the non-IID setting, the authors should compare their approach with other solutions in the IID scenario. If doable, to strengthen their contribution, the authors should compare their solution against another solution that is designed to handle the non-IID setting.*

AR: Thank you for this advice. We have evaluated the performance of FD methods on the benchmark datasets where clients’ datasets follow the IID distribution. As shown in Table 8 (Table 1 in the manuscript), all the methods achieve satisfactory performance in the IID scenario. In the revised manuscript, we have added these results within the “Performance Evaluation” subsection.

MNIST	Fashion MNIST	CIFAR-10
Conv(10,5,0)	Linear(1024)	ResNet*-18
ReLU	ReLU	Linear(128)
MaxPool(2)	Linear(1024)	ReLU
Conv(20,5,0)	ReLU	Linear(64)
ReLU	Linear(10)	ReLU
MaxPool(2)		Linear(10)
Linear(50)		
ReLU		
Linear(10)		

Table 6: The architectures of the global model in FedAvg. ResNet* represents the layers of ResNet before the fully connected layer.

Figure 5: Model inversion (MI) attacks on MNIST. (a) Visualization of training and reconstructed images. (b) Peak signal-to-noise ratio (PSNR) of reconstructed images. The error bar represents the standard deviation across 100 reconstructed images.

To the best of the authors’ knowledge, there is currently no established research specifically addressing the non-IID problem in federated knowledge distillation. Improving the performance of FD in non-IID settings is a promising avenue for future research endeavors.

Comment 2.3

RC: *The authors should introduce the thresholds and their purpose at the beginning of the “Threshold Analysis”. The thresholds are only introduced in Methods afterward.*

AR: Thank you for this advice. In the revision, we have provided more details to describe the thresholds τ_{client} and τ_{server} in this subsection.

Communication overhead	Selective-FD			FedAvg	
	Collect proxy samples	Download	Upload	Download	Upload
MNIST	4.7 MB	21.5 KB	20.5 KB	87.4 KB	87.4 KB
Fashion MNIST	4.7 MB	21.5 KB	20.5 KB	7.46 MB	7.46 MB
CIFAR-10	30.7 MB	1.3 KB	1.3 KB	45.1 MB	45.1 MB

Table 7: Communication overhead of each client. Note that collecting proxy samples only incurs a one-time cost, while downloading and uploading are needed for each iteration.

Strong Non-IID	MNIST		FashionMNIST		CIFAR-10	
	Hard label	Soft label	Hard label	Soft label	Hard label	Soft label
IndepLearn	10.00±0.00		10.00±0.00		10.00±0.00	
FedMD	18.89±0.30	88.71±0.28	16.54±0.25	64.63±0.37	10.71±0.38	15.78±1.39
FedED	11.49±0.25	11.92±0.41	12.45±0.44	12.52±0.38	11.83±0.26	12.04±0.30
DS-FL	19.72±0.32	35.25±0.36	17.54±0.11	35.98±0.43	10.87±0.25	12.07±0.32
FKD	10.00±0.00	10.00±0.00	10.00±0.00	10.00±0.00	10.00±0.00	10.00±0.00
PLS	10.00±0.00	10.00±0.00	10.00±0.00	10.00±0.00	10.00±0.00	10.00±0.00
Selective-FD	85.92±0.37	94.68±0.52	73.41±0.98	75.31±0.29	80.22±0.74	80.98±0.39
Weak Non-IID	MNIST		FashionMNIST		CIFAR-10	
	Hard label	Soft label	Hard label	Soft label	Hard label	Soft label
IndepLearn	19.96±0.00		19.82±0.01		19.52±0.02	
FedMD	26.77±0.57	95.16±0.52	41.92±0.30	74.83±0.41	62.14±0.22	84.31±0.53
FedED	59.95±1.11	60.26±1.80	32.62±1.09	37.12±0.85	53.11±0.34	56.13±0.14
DS-FL	25.53±1.43	47.87±0.31	23.08±0.23	39.22±0.26	33.22±0.54	52.51±0.70
FKD	19.97±0.01	19.98±0.00	19.54±0.13	19.71±0.07	19.50±0.02	19.51±0.01
PLS	19.96±0.01	19.97±0.00	19.64±0.09	19.70±0.03	19.51±0.01	19.52±0.01
Selective-FD	86.82±0.26	96.30±0.25	75.57±0.61	77.27±0.31	81.06±0.67	85.38±0.35
IID	MNIST		FashionMNIST		CIFAR-10	
	Hard label	Soft label	Hard label	Soft label	Hard label	Soft label
IndepLearn	98.18±0.04		86.07±0.07		84.06±0.03	
FedMD	98.59±0.04	98.63±0.01	86.88±0.02	87.25±0.02	86.02±0.09	86.31±0.06
FedED	98.20±0.11	98.26±0.06	86.83±0.14	86.88±0.04	86.54±0.07	86.87±0.12
DS-FL	98.22±0.14	98.56±0.02	86.15±0.07	86.62±0.09	85.75±0.08	85.82±0.08
FKD	98.40±0.05	98.44±0.01	86.10±0.15	86.14±0.06	84.03±0.02	84.10±0.08
PLS	98.45±0.02	98.48±0.03	86.27±0.08	86.52±0.05	84.60±0.13	84.77±0.06
Selective-FD	98.55±0.01	98.60±0.04	86.92±0.08	87.16±0.06	85.94±0.07	86.06±0.16

Table 8: Test accuracy of different methods. Each experiment is repeated five times. The results in bold indicate the best performance, while the results underlined represent the second-best performance. In the non-IID settings, our Selective-FD method performs better than the baseline methods, and the accuracy gain is more significant when using hard labels in knowledge distillation than soft labels. In the IID scenario, all the methods achieve satisfactory accuracy.

REVIEWER COMMENTS

Reviewer #1 (Remarks to the Author):

Thanks for the author's response. Most of my concerns have been addressed. I only have two remaining questions in below:

Comment 1.5: Thank the authors for providing the accuracy and communication cost. It is promising to expand the experiments to some transformer-based architecture to further verify the advantages of communication cost saving.

Comment 1.6: Thanks for the explanations, and I agree with the point that the extreme one-class scenario will lead to overfitting, which raises concerns in the medical domain. However, as the paper emphasizes non-iid data, the use of uniform testing violates the assumption that client data are non-iid. It is necessary to give a better experimental evaluation. For example, suppose several hospitals with non-iid data do the FL. In that case, the local data is imbalanced, and it is very challenging to collect data and build global testing data. How to perform the evaluation is an important question. The generalization ability could be one useful metric, but it cannot measure the internal FL model error.

Reviewer #2 (Remarks to the Author):

The authors addressed my comments and provided new interesting insights. I find the comparison in terms of privacy leakage between FL and FD helpful and interesting. For completeness, the authors should mention the distribution inference issue (mentioned by reviewer 1) in the privacy leakage section. They should also explain in a few words why the last sentence of this section is true and why it is helpful to know, e.g., depending on how privacy-sensitive an application is, only hard labels could be used?

Authors' Response to Reviews of

Selective Knowledge Sharing for Privacy-Preserving Federated Distillation without A Good Teacher

Jiawei Shao, Fangzhao Wu, and Jun Zhang
Nature Communications,

RC: Reviewers' Comment, AR: Authors' Response, Manuscript Text

Overview

We are very grateful to the reviewers for their constructive comments in this round. We are also glad to see that the concerns from previous rounds have been addressed. We have carefully revised the manuscript, and the changes made in the revision are highlighted in blue. The major change is summarized below:

- As suggested by Reviewer 1, we have incorporated the transformer-based architecture into the experiments to better verify the advantages of our proposed method.
- To address Reviewer 1's concern about the experimental evaluation, we have reported the local test accuracy on the pneumonia detection task.
- As suggested by Reviewer 2, we have provided more discussion in the privacy leakage section.

All the comments raised by the reviewers have been considered in the revision of our manuscript. The following response letter addresses all the comments in detail.

Thank you very much for dedicating your time and effort towards enhancing the quality of our paper. We look forward to hearing from you again.

Reviewer 1

Comment 1.1

RC: *Thank the authors for providing the accuracy and communication cost. It is promising to expand the experiments to some transformer-based architecture to further verify the advantages of communication cost saving.*

AR: Following the reviewers' suggestion, we extend the experiments to the transformer-based architecture to further verify the advantages of communication cost saving. Specifically, we compare the performance by adopting Multilayer Perceptron (MLP) and Vision in Transformers (ViT) as the backbone for the Fashion MNIST image classification task. The structure of MLP follows the setup in the main text. In the ViT model, the input patch size is 4, the depth of the encoder is 2, the number of heads is 8, and the output dimension per head is 512. The test accuracy and communication cost during the training process are shown in Fig. 1. In line with the findings presented in the main text, our Selective-FD method exhibits lower accuracy compared to FedAvg while enhancing communication efficiency. Additionally, the ViT model incurs less communication overhead in comparison to the MLP model. This is because the fully-connected layers in an MLP have a substantial number of parameters.

In the revised version, we have included these experimental results in the Supplementary Information.

Figure 1: Test accuracy and communication cost as functions of the communication round on the Fashion MNIST classification task. The model architecture is (a) an MLP and (b) a ViT.

Comment 1.2

RC: *Thanks for the explanations, and I agree with the point that the extreme one-class scenario will lead to overfitting, which raises concerns in the medical domain. However, as the paper emphasizes non-iid data, the use of uniform testing violates the assumption that client data are non-iid. It is necessary to give a better experimental evaluation. For example, suppose several hospitals with non-iid data do the FL. In that case, the local data is imbalanced, and it is very challenging to collect data and build global testing*

Figure 2: (a) Visualization of non-iid data distribution on the pneumonia detection task. (b) Generalization ability: the test accuracy of various methods on a global test set.

Figure 3: Local test accuracy of various methods on the pneumonia detection task.

data. How to perform the evaluation is an important question. The generalization ability could be one useful metric, but it cannot measure the internal FL model error.

AR: We would like to thank the reviewer for acknowledging that the generalization ability evaluated in our experiments is a useful metric.

Following the reviewer’s suggestion, we compare the local test accuracy of various methods on the pneumonia detection task. Specifically, we use local test sets to evaluate the performance of local models. We follow the same experimental setup as the main text except that we allocate 10% of the local private data for testing. Each client shares soft labels to transfer knowledge with other clients, and the data distribution is visualized in Fig. 2(a).

As observed in Fig. 3, all the models can perform well on the test datasets of Clients 3 and 4. This is because the local datasets of these two clients only contain one class of images. Besides, the FedED method has the worst performance since it trains only one global model through knowledge distillation, which cannot fit well for heterogeneous local datasets. In addition, IndepLearn achieves satisfactory local test accuracy. This is because the local models do not need to predict the labels of out-of-class images, and thus the prediction task of each client becomes easy. However, IndepLearn fails to generalize well on unseen data as observed in Fig. 2(b). Our Selective-FD method achieves comparable or better local test accuracy compared with the baseline methods since it can effectively share knowledge among the clients and thus overcome the data heterogeneity.

Moreover, our Selective-FD method demonstrates the best generalization ability as shown in Fig. 2(b). This is attributed to the selective sharing mechanism, which enables the model to learn more robust and generalizable features from the clients' knowledge.

Finally, we would like to once again thank the reviewer for the comment regarding the evaluation in non-iid settings. We agree that it is more practical to test local models on their own datasets than on a global dataset. However, in the field of deep learning, it is crucial to let the model generalize from the training set to unseen data. The ability to generalize is what truly makes a model useful, as it enables it to predict outputs or make decisions based on new inputs that it has not encountered during training. Therefore, the motivation behind this work is to make knowledge sharing more effective during federated training to achieve better generalization ability. Evaluating the model performance on local test accuracy may lie in the field of personalized federated learning. This is a promising avenue that offers the potential to tailor AI models to individual users, but it is beyond the scope of this work.

In the revision, we have reported these experimental results in the Supplementary Information.

Reviewer 2

RC: *The authors addressed my comments and provided new interesting insights. I find the comparison in terms of privacy leakage between FL and FD helpful and interesting. For completeness, the authors should mention the distribution inference issue (mentioned by reviewer 1) in the privacy leakage section. They should also explain in a few words why the last sentence of this section is true and why it is helpful to know, e.g., depending on how privacy-sensitive an application is, only hard labels could be used.*

AR: We are glad to see that the comments raised by the reviewer have been addressed. Following the reviewer’s suggestion, we have provided more discussion in the privacy leakage section. The details are as follows:

As shown in Fig. 4(b) (Fig. 8(b) in the manuscript), compared with sharing soft labels in Selective-FD, the reconstructed images inferred from the hard labels have a lower PSNR value. This demonstrates that sharing hard labels in Selective-FD exposes less private information than sharing soft labels. This result is consistent with Hinton’s analysis in [Ref-2-1], where the soft labels provide more information per training case. In federated training where the local data are privacy-sensitive, such as large genomic datasets [Ref-2-2], it becomes crucial to share hard labels rather than soft labels. This serves as a protective measure against potential membership inference attacks [Ref-2-3]. Notably, although the knowledge sharing methods provide stronger privacy guarantees compared with FedAvg, the malicious attackers can still infer the label distribution of clients from the shared information. Developing a privacy-enhancing federated training scheme is a promising but challenging direction for future research.

[Ref-2-1] Hinton, Geoffrey, Oriol Vinyals, and Jeff Dean. Distilling the knowledge in a neural network, *arXiv preprint arXiv:1503.02531* (2015).

[Ref-2-2] Venkatesaramani, Rajagopal, et al. Defending against membership inference attacks on Beacon services. *ACM Transactions on Privacy and Security* (2023).

[Ref-2-3] Shokri, Reza, et al. Membership inference attacks against machine learning models. in *IEEE symposium on security and privacy (SP)* (2017).

Figure 4: Model inversion (MI) attacks on MNIST. (a) Visualization of training and reconstructed images. (b) Peak signal-to-noise ratio (PSNR) of reconstructed images. The error bar represents the standard deviation across 100 reconstructed images. Higher PSNR represents better image quality.

REVIEWERS' COMMENTS

Reviewer #1 (Remarks to the Author):

Many thanks for the authors detailed response, my concerns have been addressed, I have no more questions.